# Systematic protein–protein interaction mapping for clinically relevant human GPCRs

Kate Sokolina[1,†], Saranya Kittanakom[1,†], Jamie Snider[1,†], Max Kotlyar[2], Pascal Maurice[3,4,5,6] (iD), Jorge Gandía[7,8] (iD), Abla Benleulmi-Chaachoua[3,4,5], Kenjiro Tadagaki[3,4,5], Atsuro Oishi[3,4,5], Victoria Wong[1], Ramy H Malty[9], Viktor Deineko[9], Hiroyuki Aoki[9], Shahreen Amin[9], Zhong Yao[1], Xavier Morató[7,8], David Otasek[2], Hiroyuki Kobayashi[10], Javier Menendez[1], Daniel Auerbach[11], Stephane Angers[12], Natasa Pržulj[13], Michel Bouvier[10], Mohan Babu[9], Francisco Ciruela[7,8], Ralf Jockers[3,4,5], Igor Jurisica[2,14,15] (iD) & Igor Stagljar[1,16,17,*] (iD)

## Abstract

G-protein-coupled receptors (GPCRs) are the largest family of integral membrane receptors with key roles in regulating signaling pathways targeted by therapeutics, but are difficult to study using existing proteomics technologies due to their complex biochemical features. To obtain a global view of GPCR-mediated signaling and to identify novel components of their pathways, we used a modified membrane yeast two-hybrid (MYTH) approach and identified interacting partners for 48 selected full-length human ligand-unoccupied GPCRs in their native membrane environment. The resulting GPCR interactome connects 686 proteins by 987 unique interactions, including 299 membrane proteins involved in a diverse range of cellular functions. To demonstrate the biological relevance of the GPCR interactome, we validated novel interactions of the GPR37, serotonin 5-HT4d, and adenosine ADORA2A receptors. Our data represent the first large-scale interactome mapping for human GPCRs and provide a valuable resource for the analysis of signaling pathways involving this druggable family of integral membrane proteins.

**Keywords** G-protein-coupled receptors; high-throughput screening integrative computational biology; interactome; protein–protein interactions; split-ubiquitin membrane yeast two-hybrid assay
**Subject Categories** Network Biology; Signal Transduction
**Mol Syst Biol.** (2017) 13: 918

## Introduction

G-protein-coupled receptors (GPCRs) are seven-transmembrane proteins involved in many signal transduction pathways and in numerous human diseases such as schizophrenia (Moreno *et al*, 2009), Parkinson's disease (Pinna *et al*, 2005; Dusonchet *et al*, 2009; Gandía *et al*, 2013), hypertension (Brinks & Eckhart, 2010), obesity (Insel *et al*, 2007), and multiple cancers (Lappano & Maggiolini, 2011). GPCRs propagate ligand-specific intracellular signaling cascades in response to extracellular stimuli—following ligand activation, GPCRs catalyze the exchange of GDP for GTP on the Gα subunit, leading to a decreased affinity of Gα for Gβγ. The

1 Donnelly Centre, University of Toronto, Toronto, ON, Canada
2 Princess Margaret Cancer Centre, University Health Network, University of Toronto, Toronto, ON, Canada
3 Inserm, U1016, Institut Cochin, Paris, France
4 CNRS UMR 8104, Paris, France
5 Sorbonne Paris Cité, University of Paris Descartes, Paris, France
6 UMR CNRS 7369 Matrice Extracellulaire et Dynamique Cellulaire (MEDyC), Université de Reims Champagne Ardenne (URCA), UFR Sciences Exactes et Naturelles, Reims, France
7 Unitat de Farmacologia, Departament de Patologia i Terapèutica Experimental, Facultat de Medicina, IDIBELL, Universitat de Barcelona, L'Hospitalet de Llobregat, Barcelona, Spain
8 Institut de Neurociències, Universitat de Barcelona, Barcelona, Spain
9 Department of Biochemistry, Research and Innovation Centre, University of Regina, Regina, SK, Canada
10 Department of Biochemistry, Institute for Research in Immunology & Cancer, Université de Montréal, Montréal, QC, Canada
11 Dualsystems Biotech AG, Schlieren, Switzerland
12 Department of Pharmaceutical Sciences, Leslie Dan Faculty of Pharmacy and Department of Biochemistry, Faculty of Medicine, University of Toronto, Toronto, ON, Canada
13 Department of Computing, University College London, London, UK
14 Departments of Medical Biophysics and Computer Science, University of Toronto, Toronto, ON, Canada
15 Institute of Neuroimmunology, Slovak Academy of Sciences, Bratislava, Slovakia
16 Department of Molecular Genetics, University of Toronto, Toronto, ON, Canada
17 Department of Biochemistry, University of Toronto, Toronto, ON, Canada
*Corresponding author. Tel: +1 416 946 7828; E-mail: igor.stagljar@utoronto.ca
†These authors contributed equally to this work

resulting dissociation of the hetero-trimer allows the GTP-bound Gα and free Gβγ to interact with several downstream effectors, including adenylyl cyclases, phosphodiesterases, phospholipases, tyrosine kinases, and ion channels (Dupré *et al*, 2009; Ritter & Hall, 2009).

Due to their involvement in signal transmission, GPCRs are highly druggable targets for numerous pharmaceutical compounds used for various clinical indications (Lagerström & Schiöth, 2008). To design successful treatments for these diseases, it is essential to increase the depth and breadth of our understanding of the molecular events occurring during GPCR-mediated signal transduction, and to identify all of the proteins interacting with a particular GPCR relevant for human health.

Over the last decade, numerous biochemical, cell biological, and genetic assays have been used to identify and characterize GPCR-interacting partners (Daulat *et al*, 2009; Maurice *et al*, 2011). These studies showed that, in addition to G-proteins, GPCRs also interact with a wide variety of integral membrane proteins (e.g. other GPCRs, ion channels, transporters, and other family receptors) and cytosolic proteins (e.g. arrestins, GPCR kinases, Src homology 2 and 3 (SH2− and SH3−), and PDZ-domain containing proteins; Ritter & Hall, 2009; Marin *et al*, 2012; Hall & Lefkowitz, 2014). Despite wide usage of biochemical assays such as co-immunoprecipitation (co-IP), pull-down- and affinity purification-linked to mass spectrometry (AP-MS), and protein microarrays to identify GPCR-associated proteins (Daulat *et al*, 2007, 2011; Maurice *et al*, 2008; Chung *et al*, 2013; Benleulmi-Chaachoua *et al*, 2016), these methods have not been widely applied to assay GPCR-related protein–protein interactions (PPIs) in a systematic manner on a large scale. Furthermore, these methods are technically difficult and time-consuming, involving harsh treatments for cell disruption and membrane protein solubilization, and often require optimization for each target protein complex examined (Chung *et al*, 2013; Snider *et al*, 2015).

Technical progress has also been made in developing methods based on fluorescence or bioluminescence resonance energy transfer (FRET or BRET) to study GPCR-interacting partners in live mammalian cells with kinetics that are close to real-time (Lohse *et al*, 2012; Ayoub & Pin, 2013). Nonetheless, the analysis of GPCR interactors using BRET and FRET is not readily scalable to high-throughput screening (HTS), but is rather more suited to medium-throughput screens involving a limited number of putative hits. Aside from these biochemical and cell biological approaches, genetic methods such as the conventional yeast two-hybrid (YTH) system (Fields & Song, 1989) have been used to identify proteins interacting with the soluble domains of selected GPCRs (Gavarini *et al*, 2004; Canela *et al*, 2007; Yao *et al*, 2015). Unfortunately, while interesting, these studies are restricted to the investigation of only the soluble components of particular human GPCRs for which interacting proteins are selected in the yeast nucleus, which is an unnatural cellular compartment for identifying protein interactors of integral membrane proteins. Thus, our knowledge of the interacting proteins of human GPCRs suffers potentially serious limitations and biases due to the lack of a suitable high-throughput technology to efficiently and comprehensively characterize interacting proteins of integral membrane proteins in their native cellular and membrane environment.

In this study, we used a modified membrane yeast two-hybrid (MYTH) approach (Deribe *et al*, 2009; Snider *et al*, 2010; Mak *et al*, 2012; Usenovic *et al*, 2012; Huang *et al*, 2013; Xie *et al*, 2013), specifically tailored to identify interactors of full-length integral membrane proteins, as well as in-depth bioinformatics analysis to create and annotate an interactome for 48 selected full-length, clinically relevant human GPCRs in their ligand-unoccupied state, localized to their native plasma membrane. Using this rich GPCR-interactome resource, we then prioritized candidates by systematic computational analysis for further biological studies, and carried out functional studies of selected PPIs. The GPCR-interaction network presented here will be a crucial resource for increasing our fundamental understanding of the cellular role and regulation of this important family of integral membrane proteins, and may facilitate development of new disease treatments and a clearer understanding of drug mechanisms of action.

# Results

### Selection of human GPCRs, generation of GPCR "baits", and their functional validation

We used a modified split-ubiquitin membrane yeast two-hybrid (MYTH; Stagljar *et al*, 1998; Gisler *et al*, 2008; Deribe *et al*, 2009; Snider *et al*, 2010) assay to define the interactomes of 48 full-length, human, ligand-unoccupied GPCRs localized to the plasma membrane. Specific GPCRs were selected based upon their importance for human health, specifically their direct link to human disease. We screened 44 Class A rhodopsin-like receptors to create a representative interactome of this most abundant family of GPCRs in order to identify physical interaction partners, 2 Class B secretin-like receptors (vasoactive intestinal peptide receptor 2 and retinoic acid-induced gene 2 protein), and 2 Class F receptors (smoothened and Frizzled7; Table EV1 lists the 48 GPCRs and related human diseases). An overview of the complete MYTH workflow is presented in Fig 1.

MYTH GPCR constructs ("baits") were generated from the selected 48 full-length human GPCR ORFs. All baits were N-terminally tagged with the signal sequence of the yeast mating factor α to encourage plasma membrane localization and stable expression in yeast (Deribe *et al*, 2009), and C-terminally tagged with the C-terminal half of ubiquitin (Cub) fused to an artificial transcription factor (TF) comprised of LexA and VP16 (Fig 2A; Snider *et al*, 2010). Bait fusion proteins were tested for proper expression at the yeast plasma membrane by immunofluorescence, and for lack of self-activation via the NubG/NubI test using the non-interacting yeast plasma membrane prey protein Fur4p (Snider *et al*, 2010, 2013; Fig 2B). Functionality of GPCR baits upon addition of the MYTH tag was also demonstrated using two selected GPCR baits by measuring changes in growth rate of bait-expressing yeast in the presence and absence of an agonist (Fig 2C). In summary, all GPCR baits used in this study passed stringent validation tests ensuring they are properly expressed, localized, and are functional prior to their usage in high-throughput MYTH screens to identify protein interaction partners.

### Validation of MYTH GPCR baits using known GPCR interactions

To further confirm that the addition of the Cub-TF tag to the C-termini of GPCR proteins does not disrupt their function and that the MYTH system itself represents a suitable tool for use in the

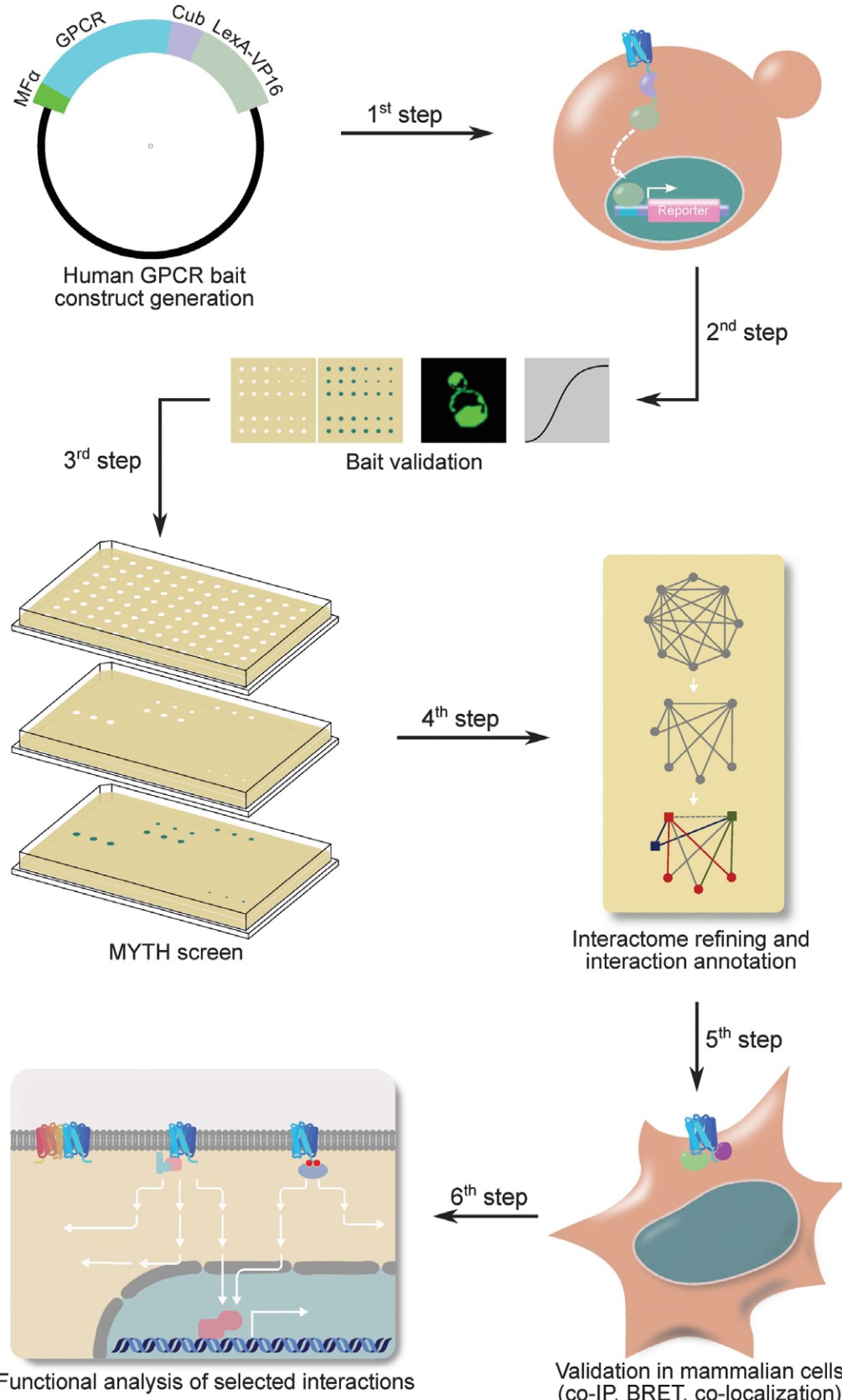

**Figure 1.  Workflow for generating the human full-length GPCR interactome.**

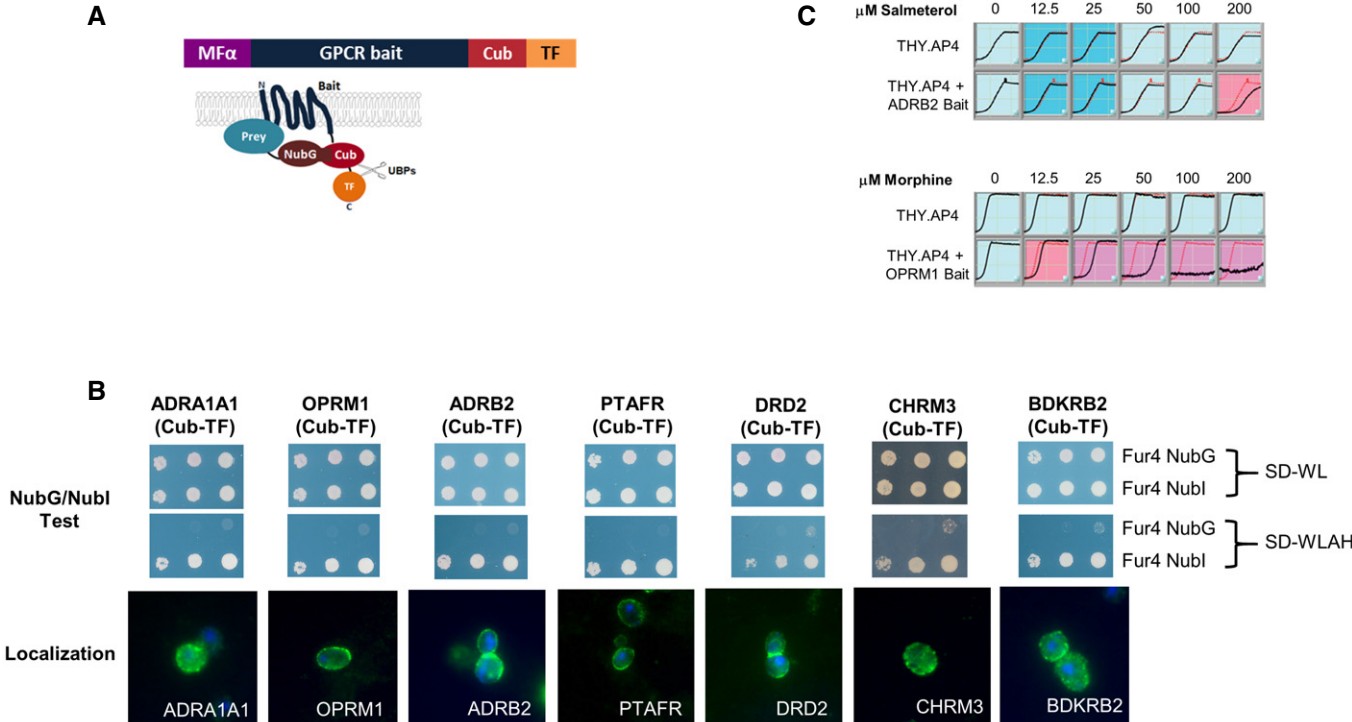

**Figure 2. Expression of human MYTH GPCR "baits" in yeast cells.**

A  The structure of the GPCR bait proteins used in this study is shown. The signal sequence of yeast α-mating pheromone precursor (MFα) was fused to the N-terminus of human GPCR baits, while the C-terminal fragment of ubiquitin (Cub) followed by an artificial transcription factor (TF) was fused to the C-terminus of the baits.

B  Representative sample of functional validation/localization tests performed on all GPCR baits used in this study. The top two panels show proper expression and MYTH function of human GPCR-Cub-TF baits demonstrated using the NubG/NubI test. In this test, GPCR-Cub-TF bait and a non-interacting yeast plasma membrane protein (Fur4p), fused to either NubI (Fur4 NubI) or NubG (Fur4 NubG) are co-expressed in yeast MYTH-reporter cells. Growth on minimal SD medium lacking Trp and Leu (SD-WL, top panel) selects only for presence of bait and prey plasmids, while minimal SD medium lacking Trp, Leu, Ade, and His (SD-WLAH, middle panel) selects for interaction between bait and prey. Co-expression of GPCR-Cub-TF bait with Fur4p fused to NubI leads to activation of the reporter system and consequent growth on SD-WLAH medium, since the wild-type NubI leads to reconstitution of ubiquitin independent of a bait–prey interaction, demonstrating that the bait protein is expressed/correctly folded. Co-expression of GPCR-Cub-TF bait and non-interacting Fur4p fused to NubG (which does not spontaneously associate with Cub) does not lead to activation of the reporter system and growth on SD-WLAH medium, demonstrating that the bait is not self-activating. The bottom panel shows localization of human GPCR bait proteins in THY.AP4 yeast reporter strain. Yeast cells expressing given human GPCR baits were fixed by paraformaldehyde and digested by zymolyase. Methanol-acetone-treated yeast spheroplasts were detected using an antibody against the transcription factor (rabbit anti-VP16) and were visualized by Cy3-conjugated secondary antibodies (shown in green). DAPI-stained nuclei can be seen as blue fluorescence. Note that similar NubGI test and localization results were obtained for all GPCR baits used in this study.

C  Growth inhibition of the human ADRB2 and OPRM1 baits expressed in yeast THY.AP4 in response to their corresponding agonist. Growth curves were carried out in triplicate, and curves shown are the average of three independent measurements at each individual time point. The red line shows the control yeast growth in the absence of drug, while the black line shows growth in the presence of drug. Inhibited growth in response to drug indicates GPCR activity.

identification of GPCR-interaction partners, we used MYTH to test a subset of 50 previously identified GPCR PPIs (Table EV2). To verify that the absence of interaction is not a false negative due to lack of prey protein expression, we made a side-by-side comparison of the NubG-tagged MYTH prey construct and the prey tagged with the original, spontaneously reconstituting wild-type NubI. Overall, 12 of the 50 (24%) could be confirmed in the MYTH assay (Fig EV1 and Table EV2). Note that not all previously reported interactions can be expected to be validated by our technique, due both to differences in the technical details of the approaches originally used (e.g. working with cell lysates instead of live cells when doing affinity purifications, working with only soluble portions of GPCRs when doing traditional YTH) and assay conditions (e.g. our assay is carried out in the absence of ligand). Our results therefore clearly demonstrate

the robustness and accuracy of the MYTH assay to detect GPCR-interacting partners.

## Building of the GPCR interactome

To systematically map interacting partners of human GPCRs, we carried out MYTH screens of the 48 selected human GPCR baits against an N-terminally NubG-tagged human cDNA library, as described previously (Snider *et al*, 2010). Briefly, yeast cells expressing MYTH baits were transformed with NubG prey pools and plated onto SD-WLAH growth media. Positive colonies were subjected to additional selection steps, and prey DNA was then isolated and sequenced to identify candidate interaction partners. The results of our extensive MYTH screens were assembled into a

"preliminary" interactome, which was further refined experimentally using the bait dependency test, which allows us to both retest each interaction (thereby demonstrating reproducibility) and identify/remove spuriously interacting preys which bind to unrelated control bait (Snider *et al*, 2010, 2013; Lam *et al*, 2015). All of the interactions that passed this secondary testing were used in subsequent bioinformatics analysis and filtering (to further identify and remove false positives/spurious interactors, including signal peptide processing and ribosomal proteins which are frequently identified "non-specific" interactors associated with general translation and trafficking processes). All remaining candidates were then assembled into our final GPCR interactome, comprising 987 unique interactions between 686 proteins, including 299 membrane proteins (Fig 3 and Table EV3). Table EV4 lists the false positives/spurious interactors removed from our final interactome.

To further investigate the biological context of the generated interactome, we analyzed its enrichment for pathways, diseases, molecular function, biological process, domains, and drug targets (see Fig 3 and Table EV5). Using pathDIP 2.5 (Rahmati *et al*, 2017), we identified significantly enriched pathways, among baits and preys including transmembrane transport of small molecules (7.0% of baits and preys, adjusted $P$ = 8.7e-8), neuroactive ligand–receptor interaction (5.0% of baits and preys, adjusted $P$ = 2.3e-6), and calcium regulation in the cardiac cell (7.7% of baits and preys, adjusted $P$ = 6.9e-6; Fig EV2A).

We investigated enrichment of diseases, functions, processes, and domains among interacting preys (Table EV5). No diseases were significantly enriched among preys, after adjusting $P$-values for multiple testing. Diseases with the lowest unadjusted $P$-values included hereditary spastic paraplegia (1.6% of preys, $P$ = 4.5e-5), schizophrenia (13.6% of preys, $P$ = 1.0e-4), and neurodegenerative disorders (6.6% of preys, $P$ = 1.0e-4; Fig EV2B). Three functions were significantly enriched: calcium ion transmembrane transporter activity (2.3% of preys, adjusted $P$ = 5.4e-3), ion channel binding (2.2% of preys, adjusted $P$ = 1.7e-2), and cation-transporting ATPase activity (2.5% of preys, adjusted $P$ = 1.9e-2). Top enriched processes included transmembrane transport (15.5% of preys, adjusted $P$ = 1.1e-3), endoplasmic reticulum calcium ion homeostasis (1.3% of preys, adjusted $P$ = 1.2e-3), and ATP hydrolysis coupled proton transport (1.3% of preys, adjusted $P$ = 2.7e-2). No domains were enriched after adjusting $P$-values for multiple testing; top domains based on unadjusted $P$-values were fatty acid hydroxylase (0.5% of preys, $P$ = 8.6e-3), V-ATPase proteolipid subunit C-like domain (0.5% of preys, $P$ = 1.1e-2), and TRAM/LAG1/CLN8 homology domain (1.0% of preys, $P$ = 1.2e-2). We also investigated whether pairs of protein domains or conserved sites (one on a bait and the other on a prey) were enriched among interacting protein pairs. Top enriched pairs (adjusted $P$ < 2.7e-12) included bait domain GPCR, rhodopsin-like, 7TM (IPR017452) paired with prey domains/sites Tetraspanin, conserved site (IPR018503), Tetraspanin, EC2 domain (IPR008952), and Marvel domain (IPR008253).

A significant number of bait GPCRs are already targeted by drugs (28 of 48 proteins, $P$ = 3.1e-8 relative to all proteins; Fig 3). These drugs comprise a variety of categories, such as histamine antagonists, antiparkinson agents, and antipruritics agents, and affect diverse organ systems, including nervous, cardiovascular, and respiratory (Fig EV3). In total, 122 baits and preys are targeted by 737 drugs. These proteins and their interactions have substantial medical and economic significance (Fig 3). Drugs that target these proteins include four of the top 100 prescribed drugs and five of the top 100 selling drugs in the United States for 2014, according to data from IMS Health, reported in Medscape (2015). These selected drugs had over 27 million prescriptions and over $14 billion in sales. Using the GPCR interactome, we can gain a more detailed understanding of how these drugs, as well as other compounds, modulate disease-related pathways.

## Orthogonal validation of MYTH-identified PPIs in mammalian cells

As a secondary validation of our GPCR interactome, a subset of PPIs selected from our interaction data was tested in mammalian cells using two distinct co-immunoprecipitation (co-IP) approaches. In the first approach, FLAG-tagged GPCR interactors were overexpressed in mammalian cells, pulled-down using anti-FLAG antibody, subjected to SDS–PAGE, transferred to membranes, and probed with commercial antibody raised against their identified endogenous GPCR-interaction partner. We tested a subset of interactions corresponding to 11 different GPCR proteins, using four MYTH-identified interacting preys and two non-interacting negative control preys for each. Of the 11 GPCR baits, five performed well in our analysis, producing no more than background signal in at least one of two negative control samples, from which we were able to confirm a total of 13 (65%) of tested interactions (Figs 4A and EV4, Table EV6). Proper expression of transiently transfected preys in these blots was checked by Western blot (Fig EV5). Of the six remaining blots, two had extremely low levels of bait expression, while four produced signal in both negative controls comparable to that in test samples, under multiple test conditions, preventing meaningful interpretation of results (Fig EV4 and Table EV6). In the second approach, an additional 14 PPIs were selected, and both immunoprecipitation and subsequent Western development were performed using native antibody directed against endogenously expressed bait and prey. Of these 14 PPIs, nine (64.2%) were successfully validated (Fig 4B and Table EV6).

As an additional orthogonal validation, we were also able to use bioluminescence resonance energy transfer (BRET; Hamdan *et al*, 2006) to confirm a small subset of eight interactions, including six not validated using either of our co-IP approaches (Table EV7).

Overall, we were able to validate a substantial number of our tested interactions (28/40, 70%) using either co-IP and/or BRET (Fig 4C), providing strong support for the robustness and quality of our MYTH-generated GPCR interactome.

## Functional analysis of novel, MYTH-identified GPCR PPIs

In an attempt to frame our GPCR interactome results in a biological context, as well as demonstrate the utility of the interactome in revealing novel interactions of biological significance, we decided to validate several novel PPIs with potential impact in neurobiology: specifically, the interactions of the hydroxytryptamine (serotonin) 5-HT4d (HTR4) receptor, a promising target for Alzheimer disease (Lezoualc'h, 2007), with both GPRIN2, a G-protein-regulated inducer of neurite outgrowth 2 that interacts with G-proteins (Chen *et al*, 1999), and the Parkinson's disease-associated receptor GPR37

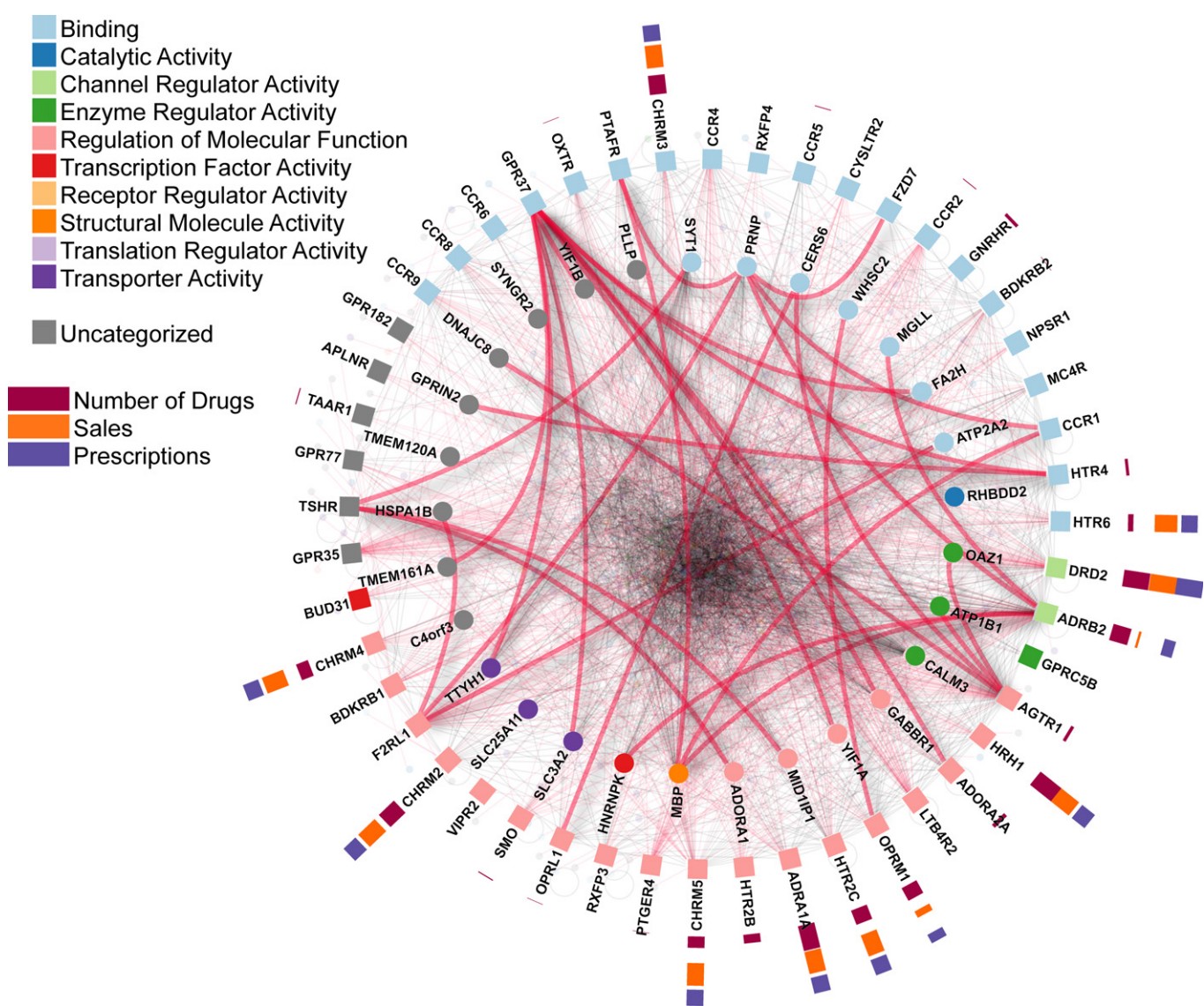

**Figure 3.  48 clinically relevant GPCR receptors mapped using MYTH.**

GPCR interactome. Validated interactions between baits–preys and preys targeted by drugs are highlighted. Drug targets were downloaded from DrugBank, drugs sales and prescription numbers were obtained from Medscape (2015). Bait–prey interactions are based on the IID database (black edges), MYTH detection (red edges), and validation assays (thick red edges). Nodes are ordered and categorized by NAViGaTOR 3's GO Molecular Function categorizer. Square nodes correspond to GPCR baits, while circular nodes correspond to interacting prey partners.

(Dusonchet *et al*, 2009), as well as the interaction between GPR37 and the adenosine A2A receptor (ADORA2A), also involved in Parkinson's disease (Pinna *et al*, 2005; Gandía *et al*, 2013).

To confirm the interaction of 5-HT4d with GPRIN2 and GPR37 in a mammalian system, we carried out co-IP experiments (Fig 5A) and BRET saturation assays (Fig 5B) in HEK-293 cells. Though the interaction with GPRIN2 is not observed by BRET, it can be detected by co-IP (Fig 5A, lanes 2 and 3), likely because the distance between *R*luc and YFP is greater than the BRET detection threshold of 100 angstroms. The interaction between 5-HT4d and GPR37 was confirmed in both assays. Co-localization of 5-HT4d with GPR37 and GPRIN2 was also observed at the plasma membrane (Fig 5C). Additionally, Erk1/2 phosphorylation and cAMP production, in

response to stimulation of 5-HT4d, were modulated by co-expressed GPR37 and GPRIN2, with ERK1/2 phosphorylation being largely abolished (Fig 5D) and maximal cAMP production potentiated (Fig 5E). This effect occurred without any modification in expression level of 5-HT4d (Fig EV6). Importantly, overexpression of GPR37 and GPRIN2 on their own did not affect cAMP production in response to agonist stimulation (Fig EV7).

Control experiments using overexpressed chemokine CCR5 receptor or a C-terminally truncated form of GPRIN2, which is unable to interact with G-proteins, did not show modulation of 5-HT4d response. GPRIN2 and GPR37 were also unable to modify the ERK and cAMP response elicited by the β2-adrenergic receptor upon isoproterenol stimulation (Fig 5D and E). Collectively, these

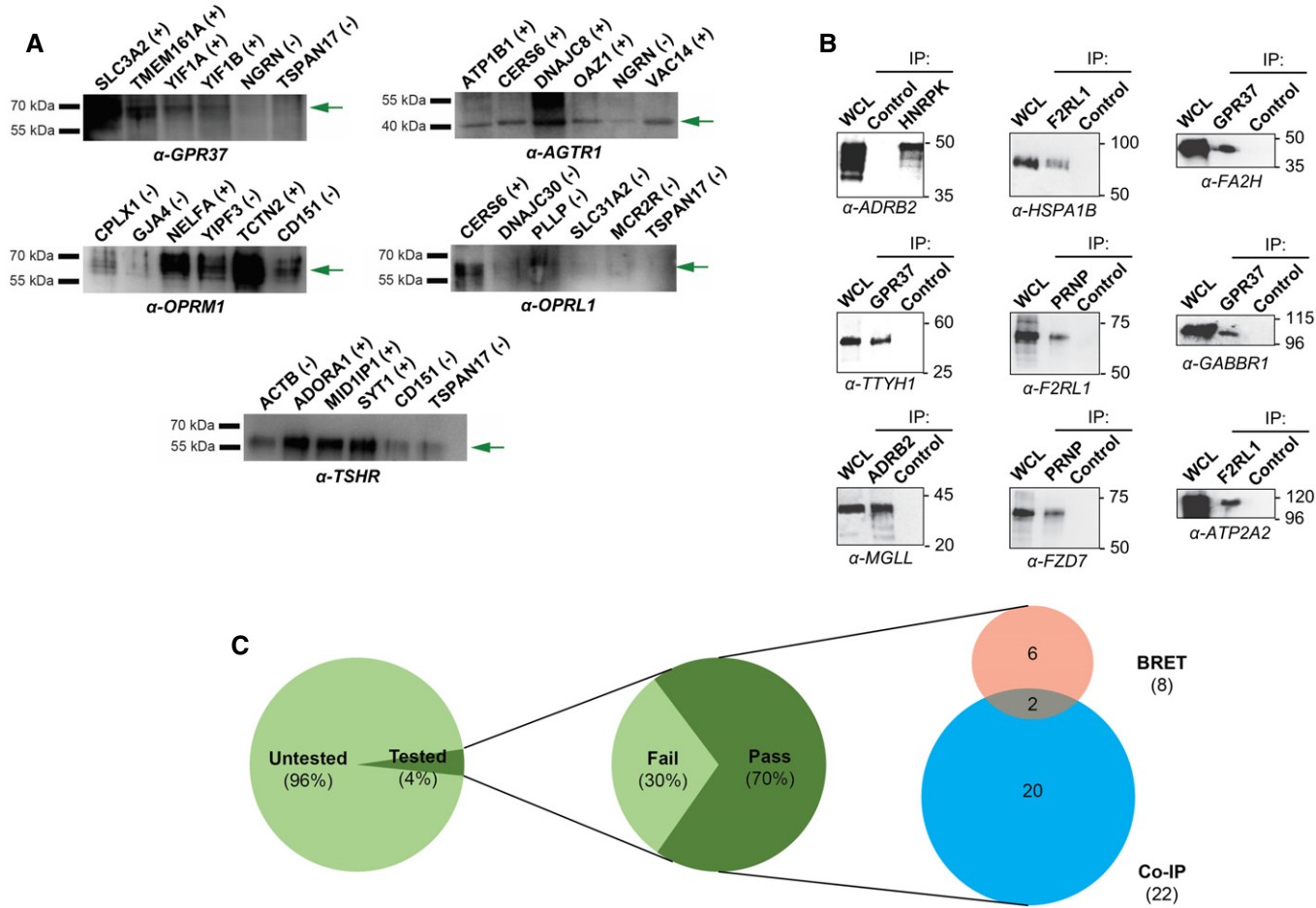

**Figure 4. Orthogonal validation of the MYTH-based GPCR interactome.**

A Co-immunoprecipitations were performed using α-FLAG antibody directed against overexpressed FLAG-tagged protein corresponding to either MYTH-identified interactor (first four lanes) or negative control (last two lanes), followed by Western blotting using antibody directed against the corresponding putative GPCR protein interaction partner (listed below each blot). All blots shown here produced no more than background signal in at least one negative control sample, making them suitable for use in validation of MYTH-detected interactions. (+) indicates an interaction was detected by co-IP. (−) indicates no interaction was detected by co-IP. Green arrows point to the band corresponding to the indicated GPCR.

B Co-immunoprecipitations were performed using native antibody directed against the interaction partner indicated below each blot, followed by Western blotting using native antibody directed against the other member of the interacting pair. All proteins were endogenously expressed. WCL, whole-cell lysate. Control, pull-down using beads only.

C A total of 40 MYTH-detected interactions were successfully tested by co-immunoprecipitation or BRET and 28 were validated, a success rate of 70%. Of the 40 interactions, 34 were tested by co-immunoprecipitation approaches and 22 of these were validated, a success rate of 64.7%. BRET was used to test eight interactions, including two tested by co-immunoprecipitation, and all were validated.

data demonstrate the specificity of the effect of GPRIN2 and GPR37 on 5-HT4d function.

Another GPCR interactor of GPR37 identified in our MYTH screen was ADORA2A, an adenosine receptor highly expressed in the striatum, a region of the brain involved in Parkinson's disease (Pinna *et al*, 2005; Gandía *et al*, 2013). The co-distribution and co-immunoprecipitation of ADORA2A and GPR37 were confirmed in HEK-293 cells (Fig 6A and B). Subsequently, the direct association between ADORA2A and GPR37 was confirmed by BRET saturation experiments (Fig 6C and D). Importantly, we did not observe a positive interaction between GPR37 and ADORA1, a related adenosine receptor (Fig 6C). Furthermore, we explored the impact of the ADORA2A/GPR37 interaction on the cell surface expression of these

receptors (Fig 6E). The levels of GPR37 when expressed alone are particularly low, as previously reported (Gandía *et al*, 2013). Interestingly, co-expression with ADORA2A markedly enhanced both whole and cell surface expression of GPR37 (Fig 6E), suggesting an ADORA2A chaperone-like function. Importantly, the expression levels of GPR37 were not enhanced by ADORA1 co-expression (Fig 6F), thus providing insight into the specificity of the ADORA2A/GPR37 interaction.

Since the levels of ADORA2A appear to affect GPR37 expression, we next aimed to explore the role of GPR37 in ADORA2A signaling *in vivo*. To this end, we first validated the ADORA2A/GPR37 interaction in native tissue, namely mouse striatum, by means of co-immunoprecipitation experiments. The immunoprecipitation of striatal

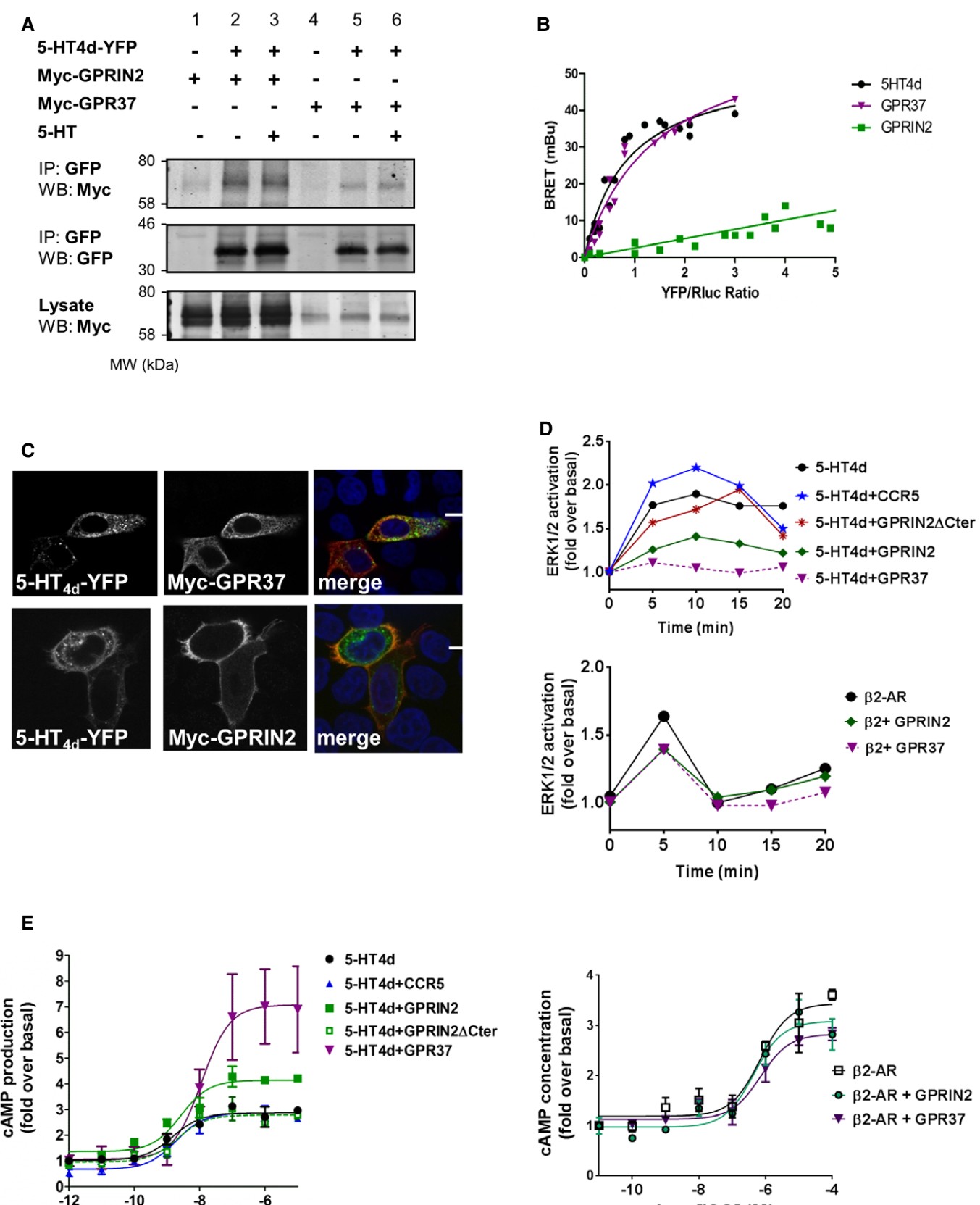

**Figure 5.**

◀

**Figure 5.  Functional interactions of GPR37 and GPRIN2 with 5-HT4d in transfected cells.**

A  Co-immunoprecipitation in the presence and absence of 1 μM 5-HT agonist for 15 min. HEK-293 cells were transiently transfected with 5-HT4d-YFP (lanes 2, 3, 5, 6) and myc-GPRIN2 (lanes 1–3) or GPR37 (lanes 4–6) and processed for immunoprecipitation using an anti-GFP antibody. The crude extracts (lysate) and immunoprecipitates (IP) were analyzed by SDS-PAGE and immunoblotted using a rabbit anti-GFP or anti-Myc antibody. Data are representative of at least two independent experiments.

B  BRET donor saturation curves were performed by co-transfecting a fixed amount of 5-HT4d-*R*luc and increasing amounts of 5-HT4d-YFP, GPR37-YFP, and GPRIN2-YFP in HEK-293 cells. Data are means of three independent experiments performed in triplicate.

C  Co-expression of HeLa cells transfected with 5-HT4d-YFP (green) and myc-GPR37 or myc-GPRIN2 (red) and analyzed by confocal microscopy. Superimposition of images (merge) reveals co-distribution in orange and DAPI-stained nuclei in blue. Scale bar: 15 μm. Data are representative of at least two independent experiments.

D  ERK1/2 activation in HEK-293 cells over time in response to 10 μM 5-HT agonist and the presence of overexpressed 5-HT4d and GPRIN2, GPR37, or CCR5. CCR5 is used as a negative control. The bottom panel shows ERK1/2 activation over time, in the presence of overexpressed $\beta_2$-adrenergic receptor and GPRIN2 or GPR37. Data are means of three independent experiments performed in triplicate.

E  Cyclic AMP levels in HEK-293 cells, in response to increasing concentrations of serotonin agonist and the presence of overexpressed 5-HT4d and GPRIN2, GPR37, or CCR5. CCR5 is used as a negative control. The right panel shows cAMP levels in response to increasing isoproterenol concentrations, in the presence of overexpressed $\beta_2$-adrenergic receptor and GPRIN2 or GPR37. Data are means of three independent experiments performed in triplicate. Error bars indicate SEM.

GPR37 yielded a band of ~45 kDa corresponding to the ADORA2A (Fig 6G). Notably, ADORA2A co-immunoprecipitation was not observed when an unrelated antibody was used, or in striatal membranes from GPR37$^{-/-}$ mice, thus validating the specificity of the interaction in native tissue. Next, we assessed the impact of GPR37 expression on ADORA2A functionality *in vivo*. Dopamine (DA) has been implicated in the central processes involved in locomotor activity (LA) regulation and psychomotor behaviors (Beninger, 1983). Interestingly, molecular and functional interactions between Dopamine Receptor 2 (D$_2$R) and ADORA2A in the nucleus accumbens are involved in mediating LA (Ferré & Fuxe, 1992). Since it appears that GPR37 interacts with both ADORA2A (from our interactome) and D$_2$R (Dunham *et al*, 2009), we assessed haloperidol-induced catalepsy in GPR37 knockout mice (GPR37$^{-/-}$) to ascertain the role of this receptor in dopamine-/adenosine-mediated psychomotor behavior. Interestingly, our results showed that in the GPR37$^{-/-}$ mice the catalepsy scores were significantly lower ($P < 0.01$) than in the GPR37$^{+/+}$ mice (Fig 6H). This result suggested a possible role of GPR37 in modulating D$_2$R-mediated neurotransmission. Next, to test the efficacy of ADORA2A in modulating haloperidol-induced catalepsy we treated animals with SCH58261, a selective A$_{2A}$R antagonist (Wardas *et al*, 2003). The administration of SCH58261 (1 mg/kg, i.p.) significantly ($P < 0.01$) reduced the catalepsy score of GPR37$^{+/+}$ animals (Fig 6H), as previously reported (Wardas *et al*, 2003). Importantly, in the GPR37$^{-/-}$ animals, SCH58261 completely abolished the haloperidol-induced catalepsia (Fig 6H). These results suggest that GPR37 might modulate D$_2$R-mediated psychomotor behavior through a putative ADORA2A/GPR37 oligomer *in vivo*.

Taken together, we were able to confirm and functionally characterize two MYTH interactions, thus further demonstrating the utility of our MYTH-based GPCR interactome as a useful resource for disease-related biological research. Annotated interactions from this study are made publicly available in the IID database (Kotlyar *et al*, 2016), with accession number #IID-003170131 (http://ophid.utoronto.ca/iid/SearchPPIs/dataset/IID-003170131).

## Discussion

Although GPCRs represent one of the most important protein classes involved in cell signaling, comprehensive studies of their interactors have been lacking because traditional high-throughput interactive proteomics assays do not make use of full-length GPCRs in a natural cellular context. In this study, we report the first systematic interactome analysis of 48 clinically important human GPCRs in their ligand-unoccupied state. We have thus created a foundational GPCR interactome, which is necessary for assessing and understanding complex signaling pathways and for elucidating mechanisms of drug action. Overall, our bioinformatics analysis of the human GPCR interactome, focusing on human diseases, provides critical and focused research directions for GPCR signaling and function.

In establishing the utility of the MYTH system to identify human GPCR interactions, we tested known GPCR-interacting proteins in MYTH, confirming 24% of tested interactions. Though not all tested interactions could be validated using MYTH, this is not unexpected due to differences in the approaches used. For instance, many of the interactions used in our test subset were previously identified using affinity purification (which makes use of cell lysates instead of live cells) or traditional YTH-based approaches (which can typically only be performed using soluble portions of membrane proteins), while MYTH allows for the study of full-length membrane proteins, directly in the membrane environment of a live cell. As such, we expect MYTH to more accurately reflect the natural cellular conditions of membrane proteins, and therefore potentially better identify membrane protein interactions, and detect fewer false positives, than traditional methods. We were still able to recapitulate a substantial percentage of previously identified interactions, however, demonstrating the effectiveness of the MYTH assay for use in the detection of GPCR interactions.

Using our MYTH screening approach, combined with comprehensive bioinformatics analysis, we were able to generate a richly annotated interactome comprised of 987 unique interactions across a total of 686 proteins. Of these, 299 were membrane proteins, demonstrating the effectiveness of MYTH in identifying membrane protein interactions. To further validate our interactome, we successfully carried our orthogonal analysis using co-IP and BRET approaches on a subset of 40 interactions spanning 10 different GPCRs, and were able to confirm a total of 28 of 40 interactions (70%). Failure to validate tested interactions, or identify conditions under which certain interactions could be properly assessed by our orthogonal methods, could be reflective of poor endogenous expression of tested GPCRs and/or aberrant interaction behavior in the unnatural and stringent environment produced upon cellular lysis. Overall, however, the strong confirmation rate obtained using our orthogonal test approaches extensively supports the quality of our MYTH GPCR interactome dataset.

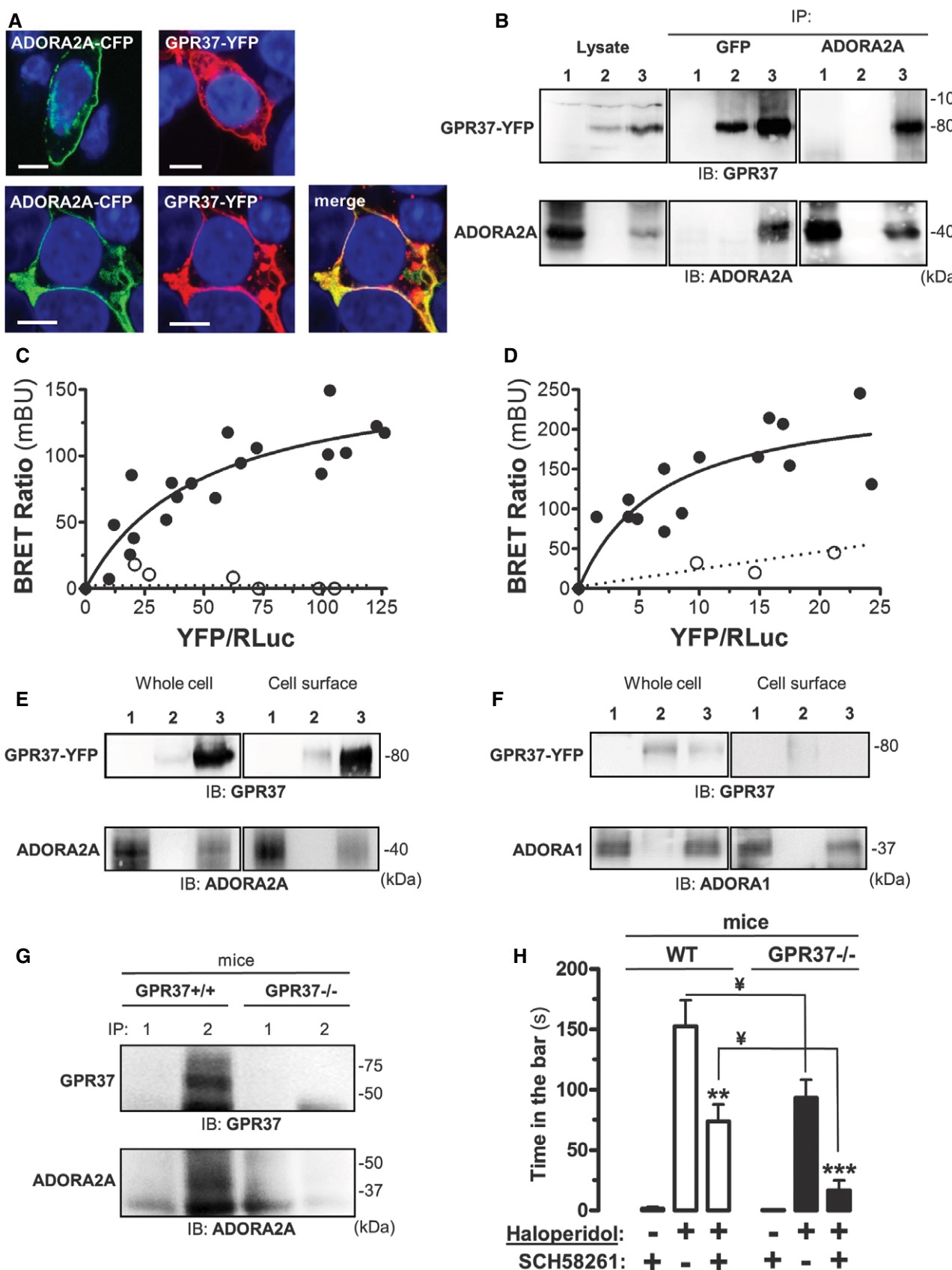

Figure 6.

◀

**Figure 6.  Validation of ADORA2A and GPR37 interaction in HEK-293 cells and native tissue.**

A    Co-localization of ADORA2A and GPR37 in HEK-293 cells transiently transfected with ADORA2A-CFP, GPR37-YFP, or ADORA2A-CFP plus GPR37-YFP. Transfected cells were analyzed by confocal microscopy. Merged images reveal co-distribution of ADORA2A-CFP and GPR37-YFP (yellow) and DAPI-stained nuclei (blue). Scale bar: 10 μm.

B    Co-immunoprecipitation of ADORA2A and GPR37 from HEK-293 transiently transfected with ADORA2A (lane 1), GPR37-YFP (lane 2) or ADORA2A plus GPR37-YFP (lane 3) using a mouse anti-GFP antibody (2 μg/ml) or a mouse anti-$A_{2A}$R antibody (1 μg/ml). The crude extracts (Lysate) and immunoprecipitates (IP) were analyzed by SDS–PAGE and immunoblotted (IB) using a rabbit anti-GPR37 (1/2,000) or rabbit anti-$A_{2A}$R antibody (1/2,000).

C, D    BRET saturation experiments between GPR37-*R*luc and ADORA2A-YFP (black circle) or ADORA1-YFP (white circle; C), or ADORA2A-*R*luc and GPR37-YFP (black circle) or CD4R-YFP control (white circle; D) in transiently transfected HEK-293. Plotted on the *x*-axis is the fluorescence value obtained from the YFP, normalized with the luminescence value of the *R*luc constructs 10 min after *h*-coelenterazine (5 μM) incubation, and on the *y*-axis the corresponding BRET ratio (×1,000). mBU, mBRET units. Data shown are from three independent experiments.

E    Cell surface expression of HEK-293 cells transiently transfected with cDNA encoding ADORA2A (lane 1), GPR37-YFP (lane 2) or ADORA2A plus GPR37-YFP (lane 3). Cell surface proteins were biotinylated and crude extracts (whole cell) and biotinylated proteins were subsequently analyzed by SDS–PAGE and immunoblotted (IB) using a rabbit anti-GPR37 antibody (1/2,000) or a rabbit anti-$A_{2A}$R antibody (1/2,000).

F    Cell surface expression of HEK-293 cells transiently transfected with cDNA encoding ADORA1 (lane 1), GPR37-YFP (lane 2), or ADORA1 plus GPR37-YFP (lane 3). Cell surface proteins were biotinylated and crude extracts (whole cell) and biotinylated proteins were subsequently analyzed by SDS-PAGE and immunoblotted (IB) using a rabbit anti-GPR37 antibody (1/2,000) or a rabbit anti-$A_1$R antibody (1/2,000).

G    Co-immunoprecipitation of ADORA2A and GPR37 from C57BL/6J wild-type (GPR37$^{+/+}$) and mutant (GPR37$^{-/-}$) mice striatum using a rabbit anti-FLAG antibody (4 μg/ml; lane 1) or a rabbit anti-GPR37 antibody (4 μg/ml; lane 2). The immunoprecipitates (IP) were analyzed by SDS-PAGE and immunoblotted (IB) using a rabbit anti-GPR37 (1/2,000) or mouse anti-$A_{2A}$R antibody (1/2,000).

H    Involvement of GPR37 in haloperidol-induced catalepsy. The influence of systemic injection of ADORA2A antagonist SCH 58261 (1 mg/kg, i.p.) on the catalepsy induced by haloperidol (1.5 mg/kg i.p.) was assessed in both WT (GPR37$^{+/+}$) and mutant (GPR37$^{-/-}$) mice as described in Materials and Methods. The data indicate the mean ± SEM (*n* = 6 per group). Asterisks denote data significantly different from the haloperidol-treated mice: **$P$ < 0.01 and ***$P$ < 0.001 by one-way ANOVA with Bonferroni multiple comparison *post hoc* test. In the GPR37$^{-/-}$ mice, the haloperidol plus SCH 58261 group were not significantly different ($P$ > 0.05) from the control (i.e. SCH 58261 alone). $^\yen P$ < 0.01 by two-way ANOVA with Bonferroni multiple comparison *post hoc* test for genotype and treatment comparisons.

We also carried out additional, in-depth functional validation on selected GPCR PPIs identified in our interactome using biochemical and cell-based assays as well as knockout and knock-in animals. First, we found that GPRIN2 and GPR37 physically and functionally interact with the 5-HT4d receptor, a promising target for Alzheimer's disease. Activation of 5-HT4d has been shown to modulate α-secretase activity, thus promoting the generation of the amyloid precursor protein (APP)α at the expense of the Alzheimer disease-associated APPβ (Thathiah & De Strooper, 2011). This effect involves the $G_s$/cAMP signaling pathway (Maillet *et al*, 2003). Based on our results, the suspected beneficial effect of 5-HT4d on Alzheimer disease development is expected to be amplified in cells co-expressing either GPRIN2 or GPR37.

Another functionally important interactor of GPR37 was ADORA2A, whose co-expression is observed to markedly enhance whole and cell surface expression of GPR37, and whose interaction with GPR37 we validated in native tissue. This interaction is particularly notable in light of a reported interaction between GPR37 and $D_2$R (Dunham *et al*, 2009). Both ADORA2A and $D_2$R are known to co-express (Fuxe *et al*, 2007) and interact (Hillion *et al*, 2002) in regions of the brain also expressing GPR37 (i.e. striatum), and are involved in mediating locomotor activity (Ferré & Fuxe, 1992; Lein *et al*, 2007). Taking our above data, together with our observations pertaining to the effects of GPR37 deletion in mice on haloperidol-induced catalepsy and previous findings that GPR37 affects ligand binding affinities of $D_2$R (Dunham *et al*, 2009), we hypothesize that the interaction between GPR37 and ADORA2A (and possibly with $D_2$R) may play a critical role in $D_2$R/ADORA2A-mediated psychomotor behavior, and thus may function as a homeostatic regulator of dopaminergic/adenosinergic transmission *in vivo*.

GPCR–GPCR heterodimerization has been widely reported (Prinster *et al*, 2005), and the resultant cross-talk and mutual regulation have been important for understanding the functionality of receptors (Fuxe *et al*, 2014), such as ADORA2A and $D_2$R in the brain (Fuxe *et al*, 2007; Ciruela *et al*, 2011). Our interactome data, in addition to functionally elucidated receptor interactions described above, report other novel GPCR–GPCR interactions for further investigation by the scientific community, highlighting the importance of large-scale GPCR screens, such as those performed here using MYTH, in identifying new PPIs of potential clinical relevance.

Interestingly, interacting partners were observed to have different effects on GPCR function; for example, GPRIN2 and GPR37 modulate 5-HT4d signaling capacity directly, most likely through an allosteric mechanism, whereas ADORA2A promotes GPR37 expression with important consequences on the well-established and relevant ADORA2A-mediated antagonism of $D_2$R function *in vivo*. These focused analyses of novel GPCR interactions further demonstrate the utility of our MYTH-based GPCR interactome as a powerful resource for biological research in this area.

In summary, we report here the largest, most comprehensive interactome study of full-length, human GPCRs carried out directly in the context of living cells. All of the data generated in this work is freely available for use by the scientific community [see the Expanded View and online in the IID database (Kotlyar *et al*, 2016)]. Additionally, we have performed preliminary functional validation of a selection of PPIs, which should serve as a starting point for further work. Our GPCR-interactome data, particularly when combined with other collaborative projects, such as the GPCR Network (Stevens *et al*, 2012) and the mapping of GPCR interaction networks performed using other recently developed technologies, such as CHIP-MYTH (Kittanakom *et al*, 2014) and the mammalian membrane two-hybrid (MaMTH; Petschnigg *et al*, 2014; Yao *et al*, 2017), will contribute significantly to our understanding of the chemistry and biology of these clinically relevant proteins, serving as an important tool to further our knowledge of cell signaling processes and helping identify novel biologically important interactions for use in the development and improvement of therapeutic strategies.

# Materials and Methods

### Full-length human bait generation

Each human GPCR was amplified by PCR and inserted by homologous recombination (Chen *et al*, 1992) in yeast into either of the two bait vectors pCCW-STE or pTMBV (Dualsystems Biotech). The primers used for the pCCW vector are 5′-CCTTTAATTAAGGCCGCC TCGGCCATCTGCAGG-3′ (forward) and 5′-CGACATGGTCGACGGT ATCGATAAGCTTGATATCAGCAGTGAGTCATTTGTACTAC-3′ (reverse). The primers used for the pTMBV4 vector are 5′-CCAGTGGC TGCAGGGCCGCCTCGGCCAAAGGCCTCCATGG-3′ (forward) and 5′-ATGTCGGGGGGGATCCCTCCAGATCAACAAAGATTG-3′ (reverse). In MYTH bait vectors, the GPCRs were fused N-terminally to the yeast mating factor alpha signal sequence to target full-length non-yeast membrane proteins to the membrane (King *et al*, 1990). At the C-terminus, the GPCR was fused in-frame with the MYTH tag consisting of a C-terminal ubiquitin (Cub) moiety and LexA-VP16 transcription factor (TF; Fields & Song, 1989; Fashena *et al*, 2000).

### Bait validation

The resulting MYTH bait constructs were tested as previously described (Snider *et al*, 2010, 2013). Briefly, the baits were transformed (Gietz & Woods, 2006) into either of the yeast reporter strains THY.AP4 or NMY51. The correct localization of modified baits to the membrane was confirmed by immunofluorescence using (rabbit) anti-VP16 (Sigma Cat# V4388; 1/200); secondary (goat) anti-(rabbit) Cy3 (Cedarlane Cat#111-165-003; 1/500)). Test MYTH was carried out with control interacting (NubI) preys to confirm functionality in MYTH, and with non-interacting (NubG) preys to verify that baits do not self-activate in the absence of interacting prey (Snider *et al*, 2010).

Functionality of select GPCR-Cub-TF baits (Pausch, 1997) was confirmed (Dowell & Brown, 2009) in either wild-type THY.AP4 or the same strain expressing a given GPCR-Cub-TF fusion. Cells were diluted from an overnight culture to an $OD_{600}$ of 0.0625 in minimum SD or SD-Leu media, respectively. The various concentrations of drugs, salmeterol (agonist for ADRB2) or morphine (agonist for OPRM1), were added to a final concentration of 200 μM. The growth rate was monitored by measuring the $OD_{600}$ every 15 min for 24 h by TECAN Sunrise plate reader.

### Confirmation of known GPCR interactions by MYTH

Known GPCR-interacting partners were identified from the Integrated Interactions Database (IID) (Kotlyar *et al*, 2016). Gateway compatible ORFs were obtained from the Human ORFeome Collection version 8.1 (Yang *et al*, 2011) and used, via the Gateway system (Life Technologies), to generate either N-terminally tagged preys in pGPR3N (Dualsystems Biotech) or C-terminally tagged preys in pGLigand (created in-house, Stagljar lab) depending on which end is available for tagging. All bait prey interaction tests were carried out using MYTH as previously described (Snider *et al*, 2010) in the NMY51 yeast reporter strain. Note that prior to use in interaction tests with GPCR baits all preys were tested for promiscuity by use of an artificial bait construct that consists of the single-pass transmembrane domain of human T-cell surface glycoprotein CD4 and the Cub-TF tag (Snider *et al*, 2010) and by use of the yeast protein RGT2.

### Membrane yeast two-hybrid (MYTH) screens

Bait containing yeast were transformed in duplicate with the human fetal brain DUALmembrane cDNA library in the NubG-x orientation (DualSystems Biotech) as previously described (Snider *et al*, 2010) and plated onto synthetic dropout minus tryptophan, leucine, adenine, and histidine (SD-Trp-Leu-Ade-His) plates with various amounts of 3-amino-1,2,4-triazole (3-AT) as assessed by the NubG/I control test for each individual bait. Transformants were picked and spotted onto SD-Trp-Leu-Ade-His plates containing 3-AT and X-Gal dissolved in *N,N*-dimethyl formamide. Blue colonies, expressing putative interacting preys, were used to inoculate overnight liquid cultures (SD-Trp) and plasmid DNA extracted. Plasmid DNA was used to transform *E. coli*, DH5alpha strain for amplification. Plasmid DNA was extracted once more and sent for sequencing as well as used in the bait dependency test to rule out spurious interactors, as described previously (Snider *et al*, 2010).

### Filtering interactions

To reduce the number of false positives, we eliminated detected interactions involving preys that carry out signal peptide processing (GO:0006465) and ribosomal contaminants (Glatter *et al*, 2009). We identified these preys using Gene Ontology (GO; Ashburner *et al*, 2000) annotations from the UniProt-GO Annotation database (Matthews *et al*, 2009; Dimmer *et al*, 2012), downloaded through the EMBL-EBI QuickGO browser (Binns *et al*, 2009; http://www. ebi.ac.uk/QuickGO/GTerm?id = GO:0006465#term = annotation), on September 10, 2016.

### Identifying previously known interactions

Overlap between detected interactions and interactions already reported in previous studies was identified using the IID database (Kotlyar *et al*, 2016) ver. 2016-03 (http://ophid.utoronto.ca/iid).

### Annotating interacting proteins: membrane localization

Baits and preys localized to the plasma membrane were identified using GO annotations from the UniProt-GO Annotation database (Dimmer *et al*, 2012), obtained through the EMBL-EBI QuickGO browser (Binns *et al*, 2009; http://www.ebi.ac.uk/QuickGO/GTe rm?id = GO:0006465#term = annotation) on August 31, 2016.

### Process annotations and enrichment analysis

Baits and preys were annotated with GO Slim process terms from the *goslim_generic* set (http://www.ebi.ac.uk/QuickGO/GMultiTe rm#tab = choose-terms; Table EV3), downloaded on August 31, 2016.

### Pathway annotations

Pathway annotations for baits and preys, as well as pathway enrichment analysis, were performed using the pathDIP database

(Rahmati *et al*, 2017) ver. 2.5 (http://ophid.utoronto.ca/pathDIP), using the setting "Extended pathway associations" with default parameters. *P*-values were FDR-corrected using the Benjamini–Hochberg method.

## Disease annotations and enrichment analysis

Disease annotations for baits and preys were downloaded from the DisGeNET database (Piñero *et al*, 2015) v4.0, on Aug. 31, 2016. Disease enrichment of preys was assessed by calculating hypergeometric *P*-values (using the human genome as the background population), and correcting for multiple testing using the Benjamini–Hochberg method.

## Molecular function and biological process annotations and enrichment analysis

Molecular function and biological process Gene Ontology annotations were downloaded from Gene Ontology Consortium (Gene Ontology Consortium, 2015) on November 30, 2016. Enrichment of preys for molecular functions was calculated using the topGO library version 2.24.0 in R version 3.3.1 (Alexa & Rahnenfuhrer, 2016). A topGOdata object was created with nodeSize = 10, and the runTest function was used with the default algorithm (weight01) and statistic = fisher. *P*-values were adjusted for multiple testing using the Benjamini–Hochberg method. Enrichment of preys for biological processes was calculated the same way.

## Domain annotation and enrichment analysis

InterPro domain annotations were obtained from UniProt release 2016_11 (Mitchell *et al*, 2015; UniProt Consortium, 2015). Domain enrichment of preys was assessed by calculating hypergeometric *P*-values (using the human proteome as the background population), and correcting for multiple testing using the Benjamini–Hochberg method.

Domain pairs enriched among interacting bait–prey pairs were identified in two steps. First, sets of co-occurring domains were identified for baits; each set comprised domains that always occurred together on baits. Similarly, sets of co-occurring domains were identified on preys. Domains that did not always co-occur with others were considered domain sets of length 1. Enrichment was subsequently calculated for pairs of domain sets—one set on baits and the other on preys. Domain sets were identified for three reasons: (i) to avoid redundant results from different domains representing the same proteins, (ii) to avoid excessive multiple testing penalties from non-independent tests, and (iii) for easier interpretation of results, since a domain set clarifies that enrichment analysis cannot distinguish between domains within the set. After domain sets were identified, *P*-values were calculated for domain set pairs using hypergeometric probability with the following parameters: $N$ = the number of possible interactions involving baits (number of baits × size of human proteome), $M$ = the number of detected interactions, $n$ = the number of possible pairings between the bait domain set and the prey domain set (number of baits with domain set × number of human proteins with prey domain set), and $m$ = number of interacting bait–prey pairs with corresponding domain sets. Adjusted *P*-values were calculated using the Benjamini–Hochberg method.

## Drug target enrichment and drug category enrichment

Drug targets and drug therapeutic categories were downloaded from DrugBank version 5 (Wishart *et al*, 2006). Enrichment of drug targets among GPCR baits was calculated as a hypergeometric *P*-value, using the following parameters: the number of human protein-coding genes in the HGNC database (Gray *et al*, 2015) (19,008), the number drug targets in DrugBank (4,333), the number of baits (48), and the number of baits that are drug targets (28).

Enrichment of therapeutic categories among baits and preys was calculated as hypergeometric *P*-values using the following parameters: the number of human protein-coding genes in the HGNC database (Gray *et al*, 2015) (19,008), the number of targets in a therapeutic category, the number of baits and preys (686), and the number of baits and preys that are targets in the category. We calculated *Q*-values (*P*-values adjusted for multiple testing) using the Benjamini–Hochberg method.

Drugs sales and prescription numbers were obtained from Medscape (2015).

## PPI predictions

Predictions were obtained using the FpClass algorithm (Kotlyar *et al*, 2015): a probabilistic method that integrates diverse PPI evidence including compatibility of protein domains, gene co-expression, and functional similarity, as well as other methods integrated in IID (version 2016-03, http://ophid.utoronto.ca/iid; Kotlyar *et al*, 2016). Resulting networks were visualized in NAViGaTOR 3.0 (http://ophid.utoronto.ca/navigator; Brown *et al*, 2009).

## Confirmation of interactions by co-immunoprecipitation

### Approach 1—Endogenous baits and transiently transfected FLAG-tagged preys

293T cells were maintained in Dulbecco's modified Eagle's medium (DMEM) containing 10% FBS, 100 U penicillin, and 100 μg/ml streptomycin (Fisher Scientific, cat# SV30010) and split at 80% confluence. To co-immunoprecipitate GPCRs with their preys, plasmids encoding FLAG-tagged preys were transiently transfected in 293T cells and their interaction with GPCR was detected using Western blotting with anti-GPCR antibodies.

Briefly, 293T cells were plated at 40% confluence overnight. On the following morning, cells were transfected using calcium phosphate $[Ca_3(PO_4)_2]$ kit ProFection from Promega (cat# E1200) following manufacturer's instructions. 70 μg of plasmid DNA was added to $CaCl_2$ and water, and the mixture was added to HEPES-buffered saline while vortexing. The mixture was incubated at room temperature for 30 min. Prior to adding to cells, the mixture was vortexed again. After 24 h post-transfection, $2 \times 150$ mm dishes of 293T cells/plasmid were harvested and the cells were washed with ice-cold PBS. After that, cells were cross-linked with 0.5 mM DSP at room temperature for 30 mins followed by quenching excessive DSP with a buffer containing 0.1 M Tris–HCl, pH 7.5, and 2 mM EDTA. Detached cells were centrifuged at 400 *g* for 10 min at 4°C. The cell pellet was lysed in RIPA buffer containing 1× protease inhibitor cocktail (Sigma Aldrich, cat# P2714) on ice for 30 min with occasional agitation. To aid lysis, cells were passed through a 21G needle 10×. Lysate was cleared by centrifugation at 16,000 *g* for

15 min at 4°C. A volume of cell lysate containing 10 mg protein was adjusted to 1 ml with RIPA containing 1× protease inhibitor cocktail and 3 μg of each anti-GPCR receptor antibody were added. The tube rotated for 1 h at 4°C followed by addition of 100 μl of μMACS protein-G magnetic microbeads (Miltenyi, cat# 130-071-101) with continued rotation for additional 4 h at 4°C. μMACS columns (Miltenyi, cat# 130-092-444) were equilibrated with RIPA 1× protease inhibitor cocktail. The microbeads suspension was passed through the columns, and the retained microbeads were washed 3× with 800 μl of RIPA 0.1% of detergents and 1× protease inhibitor cocktail followed by another 2× washes with 500 μl detergent-free RIPA containing 1× protease inhibitor cocktail only. Proteins bound to the microbeads were released by addition of 25 μl Laemmli loading buffer at 95°C 2×. Eluates were analyzed using SDS-PAGE and visualized using SuperSignal West Femto Maximum Sensitivity Substrate (Thermo Fisher, cat# 34094).

### Approach 2—Endogenous baits and preys

Ten 150-mm dishes of HEK-293 cells were harvested and centrifuged at 400 *g* for 10 min. The cell pellet was resuspended in 15 ml phosphate-buffered saline (PBS) and mixed with an equal volume of cross-linking reagent (1 mM dithiobis-succinimidyl propionate prepared in PBS). After 30-min incubation, the cross-linked cells pelleted by centrifugation at 400 *g* were lysed in IPLB (immunoprecipitation lysis buffer containing 1% digitonin and 1× protease inhibitor cocktail) for 30 min. The lysates were then centrifuged at 16,000 *g* for 15 min at 4°C. The cell lysate containing ~10 mg of protein was adjusted to 1 ml with IPLB (containing 1% digitonin and 1× protease inhibitor cocktail) and 3 μg of antibody specific to the target protein was added to the mixture. The samples were incubated with 100 μl of μMACS protein-G magnetic beads followed by 5-h gentle rotation at 4°C. The bead suspension was passed through the μMACS columns (equilibrated with IPLB containing 1% digitonin and 1× protease inhibitor), and the retained beads were washed three times with 800 μl of IPLB (0.1% digitonin and 1× protease inhibitor) followed by another two washes with 500 μl IPLB (1× protease inhibitor only). Co-purifying protein that bound to the beads was eluted by the addition of 25 μl Laemmli loading buffer at 95°C, and analyzed by SDS-PAGE and immunoblotting using protein-specific antibody.

### Antibodies used in co-immunoprecipitation experiments

Santa Cruz: OPRL1 (sc-15309), TSHR (sc-13936), OPRM1 (sc-15310), AGTR1 (sc-1173-G), PTAFR (sc-20732), C5L2 (sc-368573), HRH (sc-20633), CHRM5 (sc-9110), OXTR (sc-33209). Abcam: ADRB2 (ab36956), HNRPK (ab52600), F2RL (ab124227), TTYH1 (ab57582), PRNP (ab52604), MGLL (ab24701), ATP2A2 (ab2861), FA2H (ab54615), HSPA1B (ab79852). Cell Signaling: GABBR1 (3835). ProteinTech: GPR37 (14820-1-AP), FZD7 (16974-1-AP).

## Confirmation of interactions by BRET

To confirm select interactions using BRET as an orthogonal validation assay, GPCR interactors identified in MYTH assays were fused to GFP2, a blue-shifted variant of GFP, to act as BRET acceptor, and GPCR receptors to RLucII, a brighter *Renilla* luciferase mutant, to act as donor, then plotted as increasing BRET levels compared to GFP/Rluc, as previously described (Mercier *et al*, 2002; Loening *et al*, 2006; Breton *et al*, 2010).

## 5-HT4d experiments

### Materials

The cDNAs encoding human GPR37 and GPRIN2 were purchased from UMR cDNA Resource Center. The 5-HT4d-*R*luc, 5-HT4d-YFP, and HA-CCR5 constructs have been described elsewhere (Berthouze *et al*, 2005; Tadagaki *et al*, 2012). An N-terminally 6xMyc tagged version of GPRIN2 and GPR37 and C-terminally YFP tagged GPR37-YFP and GPRIN2-YFP fusion proteins were obtained by PCR using the Phusion High-Fidelity DNA Polymerase (Finnzymes). All constructs were inserted in the pcDNA3.1 expression vector and verified by sequencing. The C-terminally deleted GPRIN2ΔCter construct was obtained by mutagenesis by introducing a stop codon resulting in a truncated protein of 149 amino acids.

### Co-immunoprecipitation

HEK-293 cells transiently transfected with 5-HT4d-YFP and myc-GPRIN2 or GPR37 were analyzed in the presence and absence of 1 μM 5-HT for 15 min and processed for immunoprecipitation using a monoclonal anti-GFP antibody. Crude extracts and immunoprecipitates were analyzed by SDS–PAGE and immunoblotted using rabbit anti-GFP or anti-myc antibodies.

### BRET

BRET donor saturation curves were performed in HEK-293 cells by co-transfecting a fixed amount of 5-HT4d-*R*luc and increasing amounts of 5-HT4d-YFP, GPR37-YFP, and GPRIN2-YFP as described previously (Maurice *et al*, 2010).

### Fluorescence microscopy

HeLa cells expressing 5-HT4d-YFP and Myc-GPR37 or Myc-GPRIN2 were fixed, permeabilized with 0.2% Triton X-100, nuclei stained with DAPI (blue) and incubated with monoclonal anti-Myc antibody (Sigma, St Louis, MO; 2 mg/ml) and subsequently with a Cy3-coupled secondary antibody. GFP, Cy3, and DAPI labeling was observed by confocal microscopy.

### Signaling assays

ERK1/2 activation and cyclic AMP levels were determined in HEK-293 cells as described previously (Guillaume *et al*, 2008).

## ADORA2A experiments

### Materials

The cDNA encoding the human GPR37 (Unigene ID: Hs.725956; Source BioScience, Nottingham, UK) was amplified and subcloned into the HindIII/EcoRI restriction sites of the pEYFP vector (Invitrogen, Carlsbad, CA, USA) using the iProof High-Fidelity DNA polymerase (Bio-Rad, Hercules, CA, USA) and the following primers: FGPR37 (5′-CGCAAGCTTATGCGAGCCCCGG-3′) and RGPRYFP (5′-CGCGAATTCCGCAATGAGTTCCG-3′). GPR37 was also subcloned in the HindIII/KpnI restriction sites of the p*R*luc-N1 vector (Perkin–Elmer, Waltham, MA, USA) using the following primers: FGPR37 and RGP*R*Luc (5′-CGCGGTACCGCGCAATGAGTTCCG-3′).

The constructs for the human adenosine A2A receptor (namely, ADORA2A-YFP and ADORA2A-*R*luc) were obtained as previously described (Gandia *et al*, 2008) and ADORA2A-CFP was obtained by subcloning the adenosine receptor from ADORA2A-YFP into the pECFP-N1 plasmid.

A homemade rabbit anti-GPR37 polyclonal antibody (Lopes *et al*, 2015) was used. Other antibodies used were rabbit anti-A$_{2A}$R (Ciruela *et al*, 2004), mouse anti-A$_{2A}$R (05-717, Millipore, Temecula, CA, USA), rabbit anti-FLAG (F7425, Sigma) and rabbit anti-A$_1$R (PA1-041A, Affinity BioReagents, Golden, CO, USA).

C57BL/6J wild-type and GPR37$^{-/-}$ mice with a C57BL/6J genetic background (Strain Name: B6.129P2-Gpr37tm1Dgen/J; The Jackson Laboratory, Bar Harbor, ME, U.S.A.) were used. Mice were housed in standard cages with *ad libitum* access to food and water, and maintained under controlled standard conditions (12-h dark/light cycle starting at 7:30 AM, 22°C temperature and 66% humidity). The University of Barcelona Committee on Animal Use and Care approved the protocol, and the animals were housed and tested in compliance with the guidelines described in the Guide for the Care and Use of Laboratory Animals (Clark *et al*, 1997) and following the European Community, law 86/609/CCE.

### Immunocytochemistry

HEK-293 cells were transiently transfected with ADORA2A-CFP, GPR37-YFP, or ADORA2A-CFP plus GPR37-YFP using the Trans-Fectin Lipid Reagent (Bio-Rad) and following the instructions provided by the manufacturer. The cells were analyzed by confocal microscopy 48 h after transfection. Superimposition of images (merge) reveals co-distribution of ADORA2A-CFP and GPR37-YFP in yellow and DAPI-stained nuclei in blue.

### Co-immunoprecipitation

Membrane extracts from HEK-293 cells and C57BL/6J mouse striatum were obtained as described previously (Burgueño *et al*, 2003). Membranes were solubilized in ice-cold radioimmunoassay (RIPA) buffer (150 mM NaCl, 1% NP-40, 50 mm Tris, 0.5% sodium deoxycholate, and 0.1% SDS, pH 8.0) for 30 min on ice in the presence of protease inhibitor (Protease Inhibitor Cocktail Set III, Millipore, Temecula, CA, USA). The solubilized membrane extract was then centrifuged at 13,000 ×*g* for 30 min, and the supernatant was incubated overnight with constant rotation at 4°C with the indicated antibody. Then, 50 µl of a suspension of Protein A–agarose (Sigma) or TrueBlot anti-rabbit Ig IP beads (eBioscience, San Diego, CA) was added and incubated for another 2 h. The beads were washed with ice-cold RIPA buffer and immune complexes were dissociated, transferred to polyvinylidene difluoride membranes and probed with the indicated primary antibodies followed by horseradish peroxidase (HRP)-conjugated secondary antibodies. The immunoreactive bands were detected using Pierce ECL Western Blotting Substrate (Thermo Fisher Scientific) and visualized in a LAS-3000 (FujiFilm Life Science).

### BRET

For BRET saturation experiments, HEK-293 cells transiently transfected with a constant amount of cDNA encoding the *R*luc constructs and increasing amounts of YFP tagged proteins were rapidly washed twice in PBS, detached and resuspended in Hank's balanced salt solution (HBSS) buffer (137 mM NaCl, 5.4 mM KCl,

0.25 mM Na$_2$HPO$_4$, 0.44 mM KH$_2$PO$_4$, 1.3 mM CaCl$_2$, 1.0 mM MgSO$_4$, 4.2 mM NaHCO$_3$, pH 7.4), containing 10 mM glucose and processed for BRET determinations using a POLARstar Optima plate-reader (BMG Labtech, Durham, NC, USA; Ciruela *et al*, 2015) or Mithras plate reader (Berthold Technologies; Cecon *et al*, 2015).

### Cell surface expression

HEK-293 cells were transiently transfected with the cDNA encoding ADORA2A, ADORA1, GPR37-YFP, ADORA2A plus GPR37-YFP or ADORA1 plus GPR37-YFP. Cell surface labeling was performed by biotinylation experiments (Burgueño *et al*, 2003). Crude extracts and biotinylated proteins were subsequently analyzed by SDS–PAGE and immunoblotted using a rabbit anti-GPR37 antibody (1/2,000), a rabbit anti-A$_{2A}$R antibody (1/2,000), or a rabbit anti-A$_1$R antibody (1/2,000). The primary bound antibody was detected as described before.

### Catalepsy score

Catalepsy behavior was induced by the D$_2$R antagonist haloperidol (1.5 mg/kg, i.p.), as previously described (Chen *et al*, 2001). Mice used in the catalepsy test were 2-month-old males. The animals were randomly distributed among the experimental groups. Fifteen min before animals were administered either saline or SCH58261 (1 mg/kg, i.p.), an A$_{2A}$R antagonist. The cataleptic response was measured as the duration of an abnormal upright posture in which the forepaws of the mouse were placed on a horizontal wooden bar (0.6 cm of diameter) at 4.5 cm high from the floor. The latency to move at least one of the two forepaws was recorded 2 h after haloperidol administration. The test was carried out by an experimenter who was blind to the identity of treatments and the cataleptic time latency was automatically recorded and counted by an independent researcher. A cutoff time of 180 s was imposed. Catalepsy testing was performed under dim (16 lux) light conditions. The sample size was initially set as five determinations per experimental condition. Subsequently, the statistical power was calculated using the IBM SPSS Statistics (version 24) software. Accordingly, the sample size was then designed to achieve a minimum of 80% statistical power.

### Data availability

All interactome data are available in the IID database (accession: IID-003170131; http://iid.ophid.utoronto.ca/SearchPPIs/dataset/IID-003170131).

**Expanded View** for this article is available online.

### Acknowledgements

We thank K. Seuwen (Novartis) for discussions during initiation phase of the project and for providing bait cDNA. The work in the Stagljar laboratory was supported by grants from the Canadian Institutes of Health Research (CIHR, #MOP-106527), Canadian Foundation for Innovation, Natural Sciences and Engineering Research Council of Canada, Ontario Genomics Institute, Canadian Cystic Fibrosis Foundation, Canadian Cancer Society and Ontario Research Fund (University Health Network). The work in the Jockers laboratory was performed within the Département Hospitalo-Universitaire (DHU) AUToimmune and HORmonal diseaseS and supported by grants from the Institut National de la Santé et de la Recherche Médicale (INSERM), the Fondation Recherche Médicale (Equipe FRM DEQ20130326503 to R.J.), the Association

pour la Recherche sur le Cancer (ARC, SFI20121205906, to R.J.), a doctoral fellowship from the CODDIM 2009 (Région Ile-de-France to A.B.C.) and a research fellowship of the Université Paris Descartes (to K.T.) and the "Who am I?" laboratory of excellence No. ANR-11-LABX-0071 funded by the French Government through its "Investments for the Future" program operated by The French National Research Agency (ANR) under grant ANR-11-IDEX-0005-01. The Jurisica laboratory was supported by Ontario Research Fund (GL2-01-030), Canada Foundation for Innovation (CFI #12301, #203373, #29272, #225404), Canada Research Chair Program (CRC #203373 and #225404), Natural Sciences Research Council (NSERC #203475) and IBM. The work from Babu's laboratory was supported by the grants from CIHR (MOP# 132191) and Saskatchewan Health Research Foundation (SHRF #2895). The Ciruela laboratory was supported by MINECO/ISCIII (SAF2014-55700-P, PCIN-2013-019-C03-03, and PIE14/00034), the Catalan Government (2014 SGR 1054), ICREA (ICREA Academia-2010), Fundació la Marató de TV3 (Grant 20152031), and FWO (SBO-140028). Also, J.G., X.M., and F.C. belong to the "Neuropharmacology and Pain" accredited research group (Generalitat de Catalunya, 2014 SGR 1251). The Przulj laboratory was supported by the European Research Council (ERC) Starting Independent Researcher Grant 278212, the National Science Foundation (NSF) Cyber-Enabled Discovery and Innovation (CDI) grant OIA-1028394, the Serbian Ministry of Education and Science Project III44006, and ARRS Project J1-5454.

## Author contributions

IS designed the project and was involved in the writing of the manuscript, and IJ managed the bioinformatics analysis of the interactome. KS and JS compiled and managed data, were actively involved in the analysis, and wrote the bulk of the manuscript. SK# created baits, carried out screening, co-immunoprecipitation, growth curve, and co-localization experiments. VW, DA, and JM carried out bait generation, bait validation, and screening. VW was also involved in bait localization and data compilation, and, with ZY, known PPI confirmations. MK, DO, and IJ performed bioinformatic analysis and generated the interactomes. NP analyzed the structural complexity of the interactome. RHM VD, HA, and SA from the Babu laboratory carried out the co-immunoprecipitation experiments to confirm interactions, and the experiments were overseen by MBo, RJ oversaw the serotonin experiments and critically reviewed the manuscript. PM, AB-C, and AO carried out the serotonin experiments, and KT performed the serotonin BRET experiments. FC oversaw the adenosine experiments carried out by JG and XM, and critically reviewed the manuscript. MBa, SA, and HK were involved in the preparation of the interactome.

## Conflict of interest

The authors declare competing financial interests. I.S. is co-founder and D.A. was the CEO/Operations of Dualsystems Biotech, Switzerland.

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
