## [Review Process File · Molecular Systems Biology]

Systematic protein-protein interaction mapping for clinically relevant human GPCRs

Kate Sokolina, Saranya Kittanakom, Jamie Snider, Max Kotlyar, Pascal Maurice, Jorge Gandía, Abba Benleulmi-Chaachoua, Kenjiro Tadagaki, Atsuro Oishi, Victoria Wong, Ramy H. Malty, Viktor Deineko, Hiroyuki Aoki, Shahreen Amin, Zhong Yao, Xavier Morató, David Otasek, Hiroyuki Kobayashi, Javier Menendez, Daniel Auerbach, Stephane Angers, Natasa Pržulj, Michel Bouvier, Mohan Babu, Francisco Ciruela, Ralf Jockers, Igor Jurisica and Igor Stagljjar

Corresponding author: Igor Stagljjar, University of Toronto

Review timeline:

Submission date:	04 November 2016
Editorial Decision:	02 December 2016
Revision received:	04 January 2017
Editorial Decision:	31 January 2017
Accepted:	03 February 2017

Editor: Maria Polychronidou

Transaction Report:

1st Editorial Decision

02 December 2016

Thank you again for submitting your work to Molecular Systems Biology. We have now heard back from the two referees who agreed to evaluate your study. As you will see below, the reviewers appreciate that the presented GPCR interactome is a useful resource. However, they list several issues, which we would ask you to address in a revision. I think that the reviewers recommendations are rather clear so there is no need to repeat the points listed below, but please let me know in case you would like to further discuss any specific point.

REFeree REPORTS

Reviewer #1:

Summary

The paper „Systematic interactome mapping of protein-protein interactions for clinically-relevant human GPCRs " by Sokolina et al uses an enhanced membrane yeast two hybrid (MYTH) approach to describe the interactome of 48 human GPCRs in a proteome wide prey screen. First the authors select bait receptors and then very carefully set up the conditions for their MYTH interactions screen. Next the 48 GPCRs are screened for binary interactions using a human cDNA library. The resulting hits are further substantiated through additional rounds of validation experiments. The resulting MYTH interactions are assembled into a network graph and pathway enrichment analysis is performed. Selected MYTH interactions are then tested in mammalian cell systems using Co-IP and BRET assays in order to validate the results and translate findings from yeast to cell lines.

Finally, the functional consequences of a few selected validated interactions are investigated, showing that new interactions indeed effect GPCR signal transduction.

General review

This paper can be structured into four parts. First, the implementation of the MYTH screen, whose technical quality is very high and very convincing. It is clearly an extension of the authors' previous work in this field and the result is impressive. The second part attempts to integrate the rich information generated in the first part into a GPCR interaction network. Unfortunately this part is much weaker than the first, and only a few actual results are generated by this analysis.

The story of the paper has a clear trajectory, as the authors first validate their hits in mammalian cells and then proceed to show functional impacts of their interactions on signal transduction. All conclusions are supported by the data.

Studying the interactions of GPCRs has been quite challenging for technical reasons, and this highly relevant group of proteins is not well studied in this regard. The limitations are mainly related to the physicochemical difficulties in studying membrane protein interactions. The MYTH screen is well suited for this problem when compared to affinity based methods and measures have been taken to document the robustness of the claimed high confidence interactions. The impressive dataset represents the most extensive systematic GPCR interactions dataset reported so far and is certainly of high value as a resource. On the other hand, the systems level description and analysis of the dataset is somewhat rudimentary and does not lead to interesting new insights that may help us to better understand how functional diversity of GPCR signaling is potentially linked to the observed interactome. Specifically it would also be interesting to classify the GPCRs based on their interactions to see similarities and differences that may relate to functional differences. Since it can be assumed that the found protein-protein interactions are direct it would be informative to include a computational analysis to see whether the found binary interactions could at least in part be explained by the presence of known structural motifs.

The parts of the paper describing functional consequences of the reported interactions support the claim that these interactions are important leads to direct future experiments to study GPCR signaling mechanisms.

The article is of potential interest for the readership of Molecular Systems Biology specifically for those interested in cell signaling and drug development. In its current version it reads primarily as a resource and I would strongly recommend expanding the systems level analysis of the data as described above in a revised version of the manuscript. I have also included some specific points listed below that need to be addressed in a revised version.

My specific comments are:

Major points:

1. There are approximately 800 human GPCRs. Please state your rationale for selecting the 48 receptors. Also discuss possible biases through this selection process and which subgroups are more or less strongly represented.
2. Figure 3A All data concerning drug sales and prescriptions are based on the Medscape webpage. Is this peer-reviewed information? If so, maybe it is better to cite the source of the webpage here.
3. Figure 3B. I do not understand this figure. What does it show? More importantly, what do we learn from it? There are no conclusions based on it in the text. Consider removing it or detail the conclusions in the text. How exactly where the PPIs selected for this figure?
4. Figure 4A. The blots shown for C5L2 and PTAFR do not support the authors' statements. The blots do not have sufficient quality for any conclusions. Also anti Flag Western blots to document transgene expression are missing in the Co-IPs shown in 4A.
5. Page 7: I do not agree with the way the success rate for the validation is calculated. We have 11 baits with 4 preys each, giving 44 validations, assessed on 11 blots. The authors consider a validation failed if the negative controls for one blot are not OK. Don't be too hard on yourself. The

test system is not working as it should, you do not need to include it in the failed validations. But, please also remove C5L2 and PTAFR from the list of validated interactions (see 4.). Your success rate should actually be higher than 55%.

Minor points:

1. Page 6. "Table EV4 lists frequently identified spurious interactors ("frequent fliers") removed from our final interactome.". Please explain the rationale and rules for removing these proteins from the dataset. What is the threshold and how is it determined for frequent fliers filtering
2. Page 6. "Drugs that target these proteins include 4 of the top 100 prescribed drugs in the United States, totalling about 27 million prescriptions in 2014; targeting drugs also include 5 of the top 100 selling drugs in the United States, with combined sales of over \$14 billion in 2014." Reference for these numbers is missing.
3. Figure 3A. Legend. Add that squares are baits and circles are preys.
4. Legends for figure 4A and EV4. Check the descriptions. The last two lanes are not always negative controls. Bait names are below the blots.
5. Title Figure EV2: "Disease and pathway enrichment analysis across baits and preys in MaMTH GPCR interactome". MaMTH is confusing and probably a mistake since I thought the yeast MTH systems was used to infer GPCR protein interactions

Reviewer #2:

Report on Sokolina et al. „Systematic interactome mapping of protein-protein interactions for clinically-relevant human GPCRs".

The Stajlar lab is developing membrane two-hybrid systems since quite a while and here report a library screen involving 50 human GPCRs and ~ 680 interaction partners. As such they exploit a previously not well covered interactome space of membrane proteins focusing on 44 rhodopsin-like receptors plus some others and provide a valuable data set for further analyses. Bait quality is carefully controlled in the heterologous yeast system. As data processing step, there is a knowledge based filtering step essentially enriching for membrane proteins in the PPI candidate partner proteins.

Enrichment analyses reveals cellular processes and diseases that associate with the interaction partners. Figure 3 provides an overview of the drugable part of the GPCR interactome. A number of interactions is validated using orthogonal assays, notably precipitating endogenous proteins. Three interactions were monitored in more detailed including co-localization, IP and dBRET PPI analyses. Stimulation of 5-HT₄d and the effect on cAMP release and ERK phosphorylation are altered when co-transfecting interacting GPCRs. ADORA2A expression modulated the GPR37 interaction partner expression or vice versa. A selective ADORA2A antagonist shows reduced effect on the mice on the "upright on horizontal wooden bar assay" in a GPR37 ko background. As such the paper focuses in the validation part on GPR37 interacting receptors (8 interactions in their data set) and ends with a suggestive in vivo validation attempt.

This study reports on a very interesting approach to characterize the GPCR interactome. This approach provided a unique dataset, with annotation and validation that can hardly be complemented by any other of the high throughput PPI techniques known to me.

Some specific points for consideration.

*) Technical: No statistics on the sequencing readouts are provided.

*) Technical: Why do we see 40 localization panels instead of 50 in Suppl Fig 1?

*) Technical: The authors always speak about 48 baits, but report PPIS for 50, please clarify.

*) I am confused about figure 2. Two reasons: first, this is a very technical panel presenting a control experiment. There is no non-interaction control, in general terms, a panel where everything is growing is not very informative (though I believe that there is no autoactivation)! Second, these interactions are recaps from the literature and what can be learned is that there are false negatives, knowing that many more were tested. So the interactions in the panel are probably easy ones and therefore a non-interaction will be informative! E.g. combining with non-interacting preys (NUBG) in the same panel may tell something about specificity as well as true negatives are included. The experiment that focuses on retesting of literature interaction does not allow any conclusion with respect to accuracy/robustness (overstated on page 5).

*) From my point of view, it would be most interesting to see representative novel interactions (, too).

*) The data set appears quite specific or (alternative explanation) may be dominated by high false negative data. The fact that most preys interact with one receptor only could meet either explanation. Given the tools and the yeast setup, that involved prey plasmid purification and retransformation, it should be straight forward to assay a subset of preys against the whole 50 bait panel. This can help estimate the false negatives and demonstrate interaction specificity.

*) The global analysis focuses on the drugable part, so general insight on what is the interaction potential of GPCRs falls a little short. I can find e.g. four kinases in the interactome data incl. Jak1. What about signaling molecules in general. More frequently we find TEMEMs and TSPANs etc (large families of proteins, little known), which might be due to the knowledge based filtering step, but also mitochondrial respiratory chain components, NUPs, and CDKs, ... What is the expectation here?

*) Are baits excluded from the enrichment analyses? It says "enrichments in the interactome"; to provide robust analyses baits probably should be excluded.

*) Figure 3B. It includes literature data. It is unclear to what extent this figure represents the data generated in this study and what is a reflection of IID.

*) Figure 5: It remains elusive if co-expression of the two other GPCRs is actually just additive (co-expression) or reflects the interaction. The minimum control experiment is to show that expression of GPR37 and GPRIN2, each on its own, do not stimulate cAMP production/repress ERK phosphorylation. It is unclear whether the effect has anything to do with the interaction as such. The GPRIN2delta mutant rather may suggest that GPRIN2 is signaling itself; the mutant is increasing ERK1 phosphorylation and decreasing in cAMP production.

*) Though ADORA2A enhanced expression of GPR37, the levels of ADORA2A decrease in HEK293. Is there an explanation (Fig 6E)?

Responses

Reviewer #1

Comment 1: On the other hand, the systems level description and analysis of the dataset is somewhat rudimentary and does not lead to interesting new insights that may help us to better understand how functional diversity of GPCR signaling is potentially linked to the observed interactome. Specifically it would also be interesting to classify the GPCRs based on their interactions to see similarities and differences that may relate to functional differences. Since it can be assumed that the found protein-protein interactions are direct it would be informative to include a computational analysis to see whether the found binary interactions could at least in part be explained by the presence of known structural motifs.

Our Response: To determine whether detected interactions are associated with specific structural domains we searched for pairs of domains enriched among interacting protein pairs. The main results of this analysis have been added to the results section, full results have been added to Table EV5, and the analysis steps have been added to the methods section. We have also investigated whether specific domains are enriched among interacting preys.

Comment 2: There are approximately 800 human GPCRs. Please state your rationale for selecting the 48 receptors. Also discuss possible biases through this selection process and which subgroups are more or less strongly represented.

Our Response: We selected specific GPCRS, as well as the overall number for screening, based on the following criteria. First, GPCRS were chosen based on their importance for human health, specifically their direct link to human disease (as outlined in table EV1). The majority of our receptors were selected from the Class A family (as this represents the largest family accounting for ~80-85% of GPCRs), with a couple of important representative members (with human disease-related association) from Class B and Class F families, as were available to us. We have now emphasized this more clearly at the start of our results section. Additionally, the total sample size of 48 GPCRs was based largely upon what we felt could be reasonably accomplished within the timeframe of the funding we received for this GPCR-interactome project, while still providing a significant amount of new data.

Comment 3: Figure 3A All data concerning drug sales and prescriptions are based on the Medscape webpage. Is this peer-reviewed information? If so, maybe it is better to cite the source of the webpage here.

Our Response: The Medscape webpage gets sales and prescription numbers from a report issued by IMS Health, the main market intelligence company in the health industry - unfortunately, access to the original report requires a subscription to IMS Health data. The manuscript has been updated to explain that the original source of the data is IMS Health, but the citation we provide is for the Medscape article (source of the data we have used), as specified on the Medscape website.

Comment 4: Figure 3B. I do not understand this figure. What does it show? More importantly, what do we learn from it? There are no conclusions based on it in the text. Consider removing it or detail the conclusions in the text. How exactly where the PPIs selected for this figure?

Our Response: We apologize that this figure is unclear and, as we agree that it does not add substantially to our results, have removed it as recommended.

Comment 5: Figure 4A. The blots shown for C5L2 and PTAFR do not support the authors' statements. The blots do not have sufficient quality for any conclusions. Also anti Flag Western blots to document transgene expression are missing in the Co-IPs shown in 4A.

Our Response: As per this reviewer's recommendations, we have removed the C5L2 and PTAFR blots from Figure 4A which we agree, due to low expression of the GPCRs in question, are not as easy to interpret as the remaining blots. We have also now included anti-FLAG Western blots demonstrating expression of our transiently transfected interactors (Fig EV5).

Comment 6: Page 7: I do not agree with the way the success rate for the validation is calculated. We have 11 baits with 4 preys each, giving 44 validations, assessed on 11 blots. The authors consider a validation failed if the negative controls for one blot are not OK. Don't be too hard on yourself. The test system is not working as it should, you do not need to include it in the failed validations. But, please also remove C5L2 and PTAFR from the list of validated interactions (see 4.). Your success rate should actually be higher than 55%.

Our Response: We thank the reviewer for their input and have removed the results of the C5L2 and PTAFR blots from our analysis, and recalculated our success rate, as suggested.

Comment 7: Page 6. "Table EV4 lists frequently identified spurious interactors ("frequent fliers") removed from our final interactome.". Please explain the rationale and rules for removing these proteins from the dataset. What is the threshold and how is it determined for frequent fliers filtering

Our Response: False positives were identified experimentally using the 'bait dependency' test, where we retest each interaction (via MYTH) using both the original bait, and an unrelated artificial control bait, to test both reproducibility and specificity. Bioinformatics filtering was also used to identify frequent fliers, such as signal peptide processing and ribosomal proteins which are related to general translation and trafficking in the cell. We apologize that this was not explained more clearly, and have modified the section in the body of the manuscript describing the building of the GPCR interactome to better explain this.

Comment 8: Page 6. "Drugs that target these proteins include 4 of the top 100 prescribed drugs in the United States, totalling about 27 million prescriptions in 2014; targeting drugs also include 5 of the top 100 selling drugs in the United States, with combined sales of over \$14 billion in 2014." Reference for these numbers is missing.

Our Response: We've added a citation to the Medscape article reporting the numbers, and updated the text to indicate that the original source of the numbers is IMS Health.

Comment 9: Figure 3A. Legend. Add that squares are baits and circles are preys.

Our Response: A sentence explaining this has been added to the legend of Figure 3.

Comment 10: Legends for figure 4A and EV4. Check the descriptions. The last two lanes are not always negative controls. Bait names are below the blots.

Our Response: Note that in Figure 4A the last two lanes correspond to our predicted negative controls. As explained in the text, only blots which produced a positive signal in no more than one of the negative control lanes were analyzed further/used in our confirmation calculations. We apologize if this was unclear.

We have also corrected the Figure 4 legends to properly indicate that the bait names are indicated below the blots.

Comment 11: Title Figure EV2: "Disease and pathway enrichment analysis across baits and

preys in MaMTH GPCR interactome". MaMTH is confusing and probably a mistake since I thought the yeast MTH systems was used to infer GPCR protein interactions

Our Response: We apologize for the mistake (which was indeed a typographical error). We have corrected it to MYTH.

Reviewer #2:

Comment 1: Technical: No statistics on the sequencing readouts are provided.

Our Response: We apologize, but we are not clear what the reviewer would like to see here. Standard Sanger sequencing followed by BLAST analysis was used to identify sequences, and ensure that identified interactors were in frame with appropriate tags. While we could attempt to provide information on numbers of sequencing reactions performed etc. we aren't sure how this information would be of general use/interest or would help strengthen the other results presented manuscript.

Comment 2: Technical: Why do we see 40 localization panels instead of 50 in Suppl Fig 1?

Our Response: Note that in Figure EV1 we were only attempting to show a representative sample of the localization and NubGI test results performed on our baits (as presenting a complete catalogue of the results would require several figures and would be largely redundant). We acknowledge that our presentation was not terribly clear and have accordingly completely restructured the figure and included improved images in all cases. We have also moved the new figure from the Supplementary to the Main Figures as we feel that it may be more suitable within the main body of the paper (note that it is now Figure 2 instead of EV1). The previous Figure 2, which itself was significantly modified (see below) has now been moved to the Supplementary as Figure EV1.

Comment 3: Technical: The authors always speak about 48 baits, but report PPIS for 50, please clarify.

Our Response: Note that for the test of 50 known GPCR interactions (Table EV2) we were not attempting to test a known interaction for each of the 48 GPCR baits used in this study (since not all the GPCRs in fact have well-established interactions to test) but rather selected 50 known GPCR interactions (in some cases involving the same GPCR multiple times) with the goal of simply showing that, in general, the MYTH assay can be used to recapitulate a reasonable number of previously identified GPCR-interactions (and therefore is a suitable tool for use in our study). We apologize for the confusion, and have made changes to the text in an attempt to make this more clear.

Comment 4: I am confused about figure2. Two reasons: first, this is a very technical panel presenting a control experiment. There is no non-interaction control, in general terms, a panel where everything is growing is not very informative (though I believe that there is no autoactivation)! Second, these interactions are recaps from the literature and what can be learned is that there are false negatives, knowing that many more were tested. So the interactions in the panel are probably easy ones and therefore a non-interaction will be informative! E.g. combining with non-interacting preys (NUBG) in the same panel may tell something about specificity as well as true negatives are included. The experiment that focuses on retesting of literature interaction does not allow any conclusion with respect to accuracy/robustness (overstated on page 5).

Our Response: We thank the reviewer for their comments. Note that our goal with this figure was simply to demonstrate that MYTH presents a useful tool for identifying GPCR interaction candidates (and that presence of the MYTH tag does not seriously interfere with these functional interactions). While our assay did not recapitulate all previously known

interactions, it did detect a sizeable fraction, and the failure to detect certain interactions is of course not unexpected (due to technical differences in methodology, steric effects of different tags etc.); something which we do address in the manuscript.

As per the reviewer's recommendations, we have updated the figure and included two additional panels showing both the baits tested in the presence of a proper 'negative' control prey (Panel B) as well as a sampling of the results for interactions which were not successfully validated using MYTH (Panel C). Note that as this figure now shows a considerable number of additional growth assays, we felt it might be more suitable in the supplementary material, and have therefore made it the new Figure EV1 (with the original Figure EV1 now being made into Figure 2, as described above).

Comment 5: From my point of view, it would be most interesting to see representative novel interactions (, too).

Our Response: While we agree that novel interactions would be interesting, we don't feel that presenting them in this panel is consistent with what we are attempting to show at this point in the manuscript (which is still dealing with validation of the MYTH assay/baits). Rather, since the entire purpose of our MYTH screens was to identify novel interactions, we feel it is more suitable to address such interactions at that point in the manuscript. Additionally, examples of MYTH plate assay images would not be different visually than the results presented in Figure 2 (now Figure EV1), so presenting them for candidates identified in MYTH screening would unfortunately increase the size of the manuscript without really offering much in the way of new information.

Comment 6: The data set appears quite specific or (alternative explanation) may be dominated by high false negative data. The fact that most preys interact with one receptor only could meet either explanation. Given the tools and the yeast setup, that involved prey plasmid purification and retransformation, it should be straight forward to assay a subset of preys against the whole 50 bait panel. This can help estimate the false negatives and demonstrate interaction specificity.

Our Response: Note that during our MYTH screens we always attempted to make sure that our libraries were completely covered, in an effort to minimize the number of false negatives. That said, we appreciate the reviewer's comments and do acknowledge that with library screening it can be difficult to meaningfully estimate false negative rates. While the reviewer's suggestion of screening a subset of identified preys against the complete panel of 48 GPCRs is potentially useful, in fact the most informative approach would be to attempt to screen all identified preys against all GPCR baits. Unfortunately, in both cases this presents a considerable amount of additional work and expense, with uncertain results, and is something we feel extends well beyond the scope of our current manuscript. While we note that no interactome reported to date can claim to be perfect or complete, and acknowledge that ours is certainly no exception, we have done our best during our screening approach to ensure good library coverage, and feel that our GPCR interactome contains a wealth of potentially valuable information and should serve as an excellent resource in its current form.

Comment 7: The global analysis focuses on the drugable part, so general insight on what is the interaction potential of GPCRs falls a little short. I can find e.g. four kinases in the interactome data incl. Jak1. What about signaling molecules in general. More frequently we find TEMEMs and TSPANs etc (large families of proteins, little known), which might be due to the knowledge based filtering step, but also mitochondrial respiratory chain components, NUPs, and CDKs, ... What is the expectation here?

Our Response: We have added an investigation of molecular function and biological process enrichment, including mitochondrial respiratory chain components, NUPs (e.g., nucleocytoplasmic transporter activity), and CDKs (e.g., cyclin-dependent protein serine/threonine kinase activity), among interacting preys. We have updated the Results and Methods sections and have added full results of the analysis to Table EV5.

Comment 8: Are baits excluded from the enrichment analyses? It says "enrichments in the interactome"; to provide robust analyses baits probably should be excluded.

Our Response: We have repeated our disease enrichment analysis (and carried out expanded analysis as described above, in the modified text, and in the methods) excluding baits. Notably, using this approach we detected no significantly enriched diseases after adjusting for multiple testing (and have mentioned this accordingly in the manuscript). We have still provided a list of the top diseases represented based on unadjusted p-values, however (Fig EV2B).

Comment 9: Figure 3B. It includes literature data. It is unclear to what extent this figure represents the data generated in this study and what is a reflection of IID.

Our Response: We apologize that this figure was not more clear and since, after careful review, we feel that it does not add significantly to our results, we have felt it best to simply remove it.

Comment 10: Figure 5: It remains elusive if co-expression of the two other GPCRs is actually just additive (co-expression) or reflects the interaction. The minimum control experiment is to show that expression of GPR37 and GPRIN2, each on its own, do not stimulate cAMP production/repress ERK phosphorylation. It is unclear whether the effect has anything to do with the interaction as such. The GPRIN2delta mutant rather may suggest that GPRIN2 is signaling itself; the mutant is increasing ERK1 phosphorylation and decreasing in cAMP production.

Our Response: We have now included an additional supplementary figure (Figure EV6) showing that overexpression of GPRIN2 or GPR37 alone in the presence of increasing concentrations of 5-HT does not lead to an increase in cAMP production.

Comment 11: Though ADORA2A enhanced expression of GPR37, the levels of ADORA2A decrease in HEK293. Is there an explanation (Fig 6E)?

Our Response: The ADORA2A-GPR37 interaction prompted an enhanced expression of GPR37 both in total extract and at the cell surface, as the referee has pointed out. The ectopic expression of GPR37 in heterologous systems (i.e. HEK293-T cells) has been always associated with low GPR37 expression levels because of its poor folding properties. Indeed, removal of its N-terminal domain (Dunham *et al.* 2009) or part of its C-terminal tail (Gandía *et al.* 2013) favored its expression and cell surface targeting. Possibly, the interaction with ADORA2A might facilitate GPR37 folding, thus increasing its expression. Importantly, this ADORA2A-mediated chaperone-like activity is not observed for ADORA1, thus supporting the specificity of the ADORA2A-GPR37 interaction. Conversely, the ADORA2A expression is markedly down regulated when co-expressed with GPR37. A possible explanation for this could be related to a reduction in the ADORA2A folding efficacy. Indeed, while the ADORA2A-mediated chaperone-like activity will be beneficial for GPR37 expression it may reduce ADORA2A maturation when co-expressed. Currently, we are studying this 'repressor' effect of GPR37 and this will be the topic of a forthcoming manuscript.

Thank you for sending us your revised manuscript. We have now heard back from the two referees who agreed to evaluate your study. As you will see below, the reviewers are satisfied with the modifications made and think that the study is now suitable for publication.

Before we formally accept the paper, we would like to ask you to include a Data Availability section at the end of the materials and methods, describing where the interactome data have been deposited and providing the link to the dataset.

REFeree REPORTS

Reviewer #1:

All major points have been addressed in the revised manuscript. We congratulate the authors for their impressive work and support publication of the revised manuscript.

Reviewer #2:

Revised version: Sokolina et al. „Systematic interactome mapping of protein-protein interactions for clinically-relevant human GPCRs“.

In the revised version of the manuscript the authors addressed the questions raised thoroughly and further improved the manuscript, in particular Figure 2 is much more meaningful to me and shows the assay in a better logic including additional controls. In addition enrichment approaches are better controlled in the revised version.

The paper presents a unique data set addressing GPCR interactions in a high quality approach. The authors nicely demonstrate that interaction partners have direct functional consequences.

Corresponding Author Name: Igor Stagljar

Manuscript Number: MSB-16-7430R